# Gain-of-function variants in the ion channel gene *TRPM3* underlie a spectrum of neurodevelopmental disorders

Lydie Burglen[1,2†], Evelien Van Hoeymissen[3,4,5†], Leila Qebibo[1], Magalie Barth[6], Newell Belnap[7], Felix Boschann[8], Christel Depienne[9], Katrien De Clercq[3,4,5], Andrew GL Douglas[10], Mark P Fitzgerald[11], Nicola Foulds[12], Catherine Garel[1,13], Ingo Helbig[11], Katharina Held[3,4,5], Denise Horn[8], Annelies Janssen[3,4], Angela M Kaindl[14,15,16], Vinodh Narayanan[7], Christina Prager[15,16], Mailys Rupin-Mas[17], Alexandra Afenjar[1], Siyuan Zhao[18], Vincent Th Ramaekers[19], Sarah M Ruggiero[11], Simon Thomas[20], Stéphanie Valence[1,21], Lionel Van Maldergem[22,23], Tibor Rohacs[18], Diana Rodriguez[1,21], David Dyment[24], Thomas Voets[3,4]*, Joris Vriens[5]*

[1]Centre de référence des malformations et maladies congénitales du cervelet, Départementde Génétique, APHP, Sorbonne University, Paris, France; [2]Developmental Brain Disorders Laboratory, Imagine Institute, Paris, France; [3]Laboratory of Ion Channel Research, Department of cellular and molecular medicine, University of Leuven, Leuven, Belgium; [4]VIB Center for Brain & Disease Research, Leuven, Belgium; [5]Laboratory of Endometrium, Endometriosis & Reproductive Medicine, Department Development & Regeneration, University of Leuven, Leuven, Belgium; [6]Department of Genetics, University Hospital of Angers, Angers, France; [7]Translational Genomics Research Institute (TGen), Neurogenomics Division, Center for Rare Childhood Disorders, Phoenix, United States; [8]Charité – Universitäts medizin Berlin, corporate member of Freie Universität Berlin and Humboldt-Universität zu Berlin, Institute of Medical Genetics and Human Genetics, Berlin, Germany; [9]Institute of Human Genetics, University Hospital Essen, University Duisburg-Essen, Essen, Germany; [10]University Hospital Southampton NHS Foundation Trust, Southampton, United Kingdom; [11]Children's Hospital of Philadelphia, Philadelphia, United States; [12]Wessex Clinical Genetics Service, University Hospital Southampton NHS Foundation Trust, Southampton, United Kingdom; [13]Service de Radiologie Pédiatrique, Hôpital Armand-Trousseau, Médecine Sorbonne Université, Paris, France; [14]Institute of Cell Biology and Neurobiology, Charité - Universitäts medizin Berlin, Berlin, Germany; [15]Department of Pediatric Neurology, Charité - Universitäts medizin Berlin, Berlin, Germany; [16]Charité – Universitäts medizin Berlin, Center for Chronically Sick Children, Berlin, Germany; [17]Department of Neuropediatrics, University Hospital of Angers, Angers, France; [18]Department of Pharmacology, Physiology and Neuroscience, Rutgers, The State University of New Jersey, Newark, United States; [19]Division Neuropediatrics, University Hospital Liège, Liège, Belgium; [20]Wessex Regional Genetics Laboratory, Salisbury District Hospital, Salisbury, United Kingdom; [21]Sorbonne Université, Service de Neuropédiatrie, Hôpital Trousseau AP-HP, Paris, France; [22]Centre de Génétique Humaine, Université de Franche-Comté Besançon, Besancon, France; [23]Center of Clinical Investigation 1431, National Institute of Health and Medical Research, Besancon, France; [24]Children's Hospital of Eastern Ontario Research Institute, University of Ottawa, Ottawa, Canada

*For correspondence:
thomas.voets@kuleuven.be (TV);
Joris.Vriens@kuleuven.be (JV)

†These authors contributed equally to this work

Competing interest: The authors declare that no competing interests exist.

**Abstract** TRPM3 is a temperature- and neurosteroid-sensitive plasma membrane cation channel expressed in a variety of neuronal and non-neuronal cells. Recently, rare de novo variants in *TRPM3* were identified in individuals with developmental and epileptic encephalopathy, but the link between TRPM3 activity and neuronal disease remains poorly understood. We previously reported that two disease-associated variants in TRPM3 lead to a gain of channel function . Here, we report a further 10 patients carrying one of seven additional heterozygous *TRPM3* missense variants. These patients present with a broad spectrum of neurodevelopmental symptoms, including global developmental delay, intellectual disability, epilepsy, musculo-skeletal anomalies, and altered pain perception. We describe a cerebellar phenotype with ataxia or severe hypotonia, nystagmus, and cerebellar atrophy in more than half of the patients. All disease-associated variants exhibited a robust gain-of-function phenotype, characterized by increased basal activity leading to cellular calcium overload and by enhanced responses to the neurosteroid ligand pregnenolone sulfate when co-expressed with wild-type TRPM3 in mammalian cells. The antiseizure medication primidone, a known TRPM3 antagonist, reduced the increased basal activity of all mutant channels. These findings establish gain-of-function of TRPM3 as the cause of a spectrum of autosomal dominant neurodevelopmental disorders with frequent cerebellar involvement in humans and provide support for the evaluation of TRPM3 antagonists as a potential therapy.

## Editor's evaluation

This important study is of interest to scientists and clinicians working to understand genetic causes of neurodevelopmental disability, cerebellar ataxia, and epilepsy. It represents the most comprehensive functional characterization of disease-causing mutations in the TRPM3 gene and includes a discussion of novel cerebellar signs and symptoms affecting a subset of affected individuals. Using calcium imaging and electrophysiology, solid evidence is provided that disease-causing mutations have a gain-of-function phenotype when expressed together with WT subunits, as in the patients in the study who are heterozygous for the mutations.

## Introduction

TRPM3, a member of the transient receptor potential (TRP) superfamily of tetrameric ion channels, is a $Ca^{2+}$-permeable cation channel activated by increasing temperature and by ligands, including the endogenous neurosteroid pregnenolone sulfate (PS; *Vriens et al., 2011*; *Wagner et al., 2008*). The channel is best known for its role in peripheral somatosensory neurons, where it is involved in heat sensation and in the development of pathological pain (*Vriens et al., 2011*; *Su et al., 2021*; *Vangeel et al., 2020*). In addition, TRPM3 is expressed in kidney, eye, pancreas, and several regions of the CNS, such as the hippocampal formation, the choroid plexus, and the cerebellum, but little is known about the channel's physiological role in these brain tissues (*Grimm et al., 2003*; *Oberwinkler et al., 2005*; *Zamudio-Bulcock et al., 2011*).

Rare de novo nonsynonymous coding variants in neuronally expressed genes are frequent causes of global developmental delay (GDD) and intellectual disability (ID) (*Rauch et al., 2012*). Recently, de novo variants in *TRPM3* were reported in patients with developmental and epileptic encephalopathy (DEE), including 16 patients with the recurrent missense variant p.Val1002Met and two additional patients with the variants p.Pro1102Gln and p.Ser1367Thr, respectively (*de Sainte Agathe et al., 2020*; *Dyment et al., 2019*; *Gauthier et al., 2021*; *Kang et al., 2021*; *Lines et al., 2022*; see results and *Figure 1* for further discussion and details on numbering of *TRPM3* variants). These patients consistently present with moderate-to-severe GDD and ID, variably associated with other clinical features such as childhood-onset epilepsy, hypotonia, altered heat, and/or pain sensitivity and variable facial dysmorphism. Functional characterization of the p.Val1002Met and p.Pro1102Gln variants in heterologous expression systems indicated that both lead to a gain of channel function (*Van Hoeymissen et al., 2020*; *Zhao et al., 2020*). However, the relation between TRPM3 channel function and neuronal disease remains poorly understood, and it is unknown whether other variants in *TRPM3* can cause human disease.

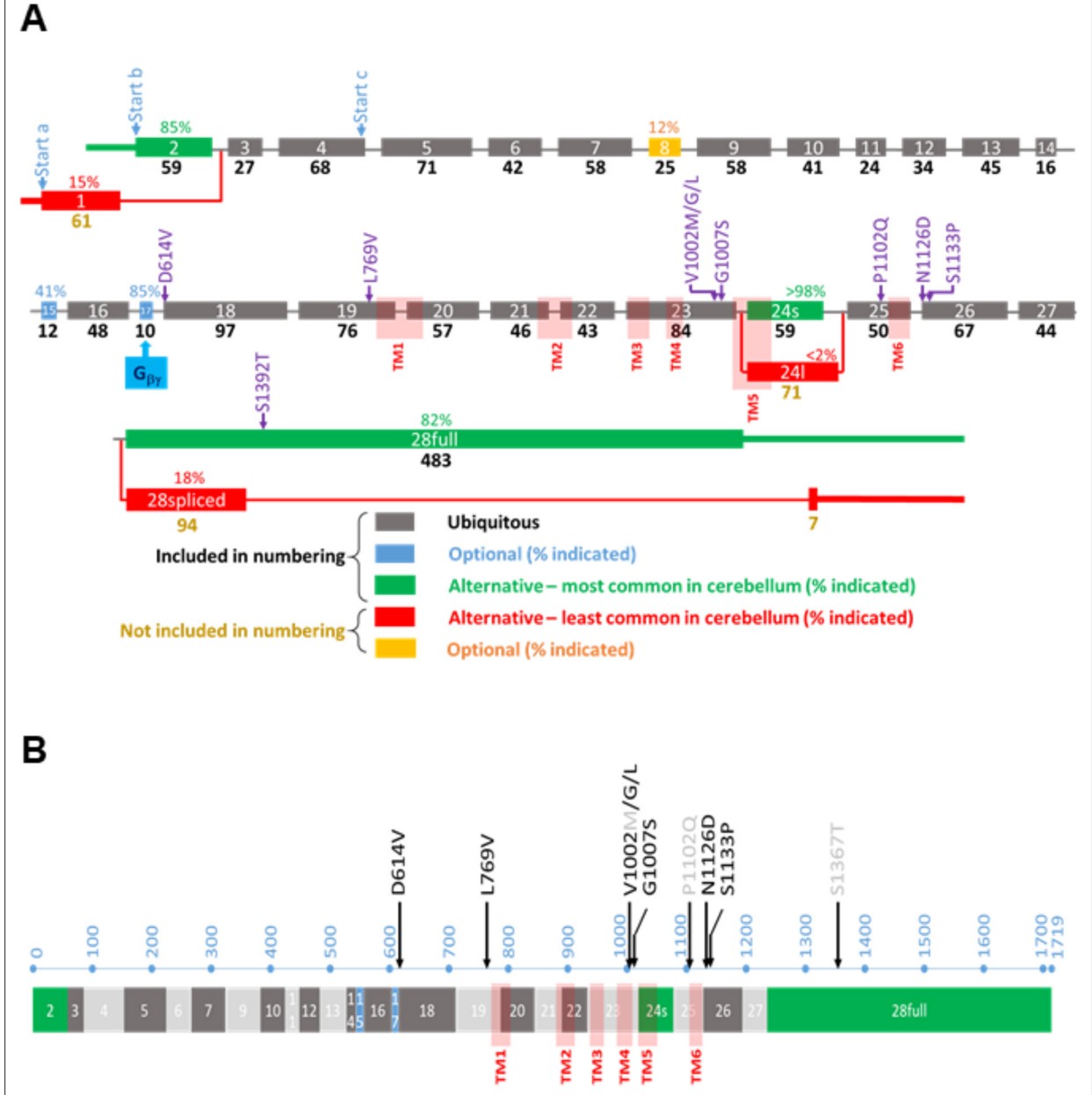

**Figure 1.** Overview of the *TRPM3* gene and location of the different variants. (**A**) Exon-intron structure and alternative splicing of *TRPM3*. Percentages above colored exons indicate the percentage of transcripts that include the indicated exons in human cerebellar RNA-seq analyses. Exons included for numbering of the disease-associated variants are indicated in gray, blue, and green, resulting in the functional channel construct indicated in (**B**). Variant numbering was based on the amino acid position of the mutated residue in the NM_001366145.2 isoform. See text for more details.

The online version of this article includes the following figure supplement(s) for figure 1:

**Figure supplement 1.** Functional characterization of different human TRPM3 constructs.

**Figure supplement 2.** Structural model of TRPM3 based on the cryo-Electron Microscopy (EM) structure (pdb: 8DDQ).

**Figure supplement 3.** Sequence alignment of TRPM3 with different species and different members of the transient receptor potential (TRP) melastatin (**M**) family.

Here, we report the clinical characteristics of 10 individuals exhibiting one of seven previously unreported heterozygous missense variants and highlight a novel cerebellar phenotype observed in more than half of the patients. Functional characterization of the newly discovered *TRPM3* missense variants in a human cell line revealed a consistent and robust increase in channel activity when co-expressed

with wild-type (WT) TRPM3 subunits. These findings establish gain-of-function variants in *TRPM3* as pathogenic, causing an autosomal dominant neurodevelopmental syndrome with frequent cerebellar involvement.

## Results

### Organization of the human *TRPM3* gene and alternative splicing in the cerebellum

The *TRPM3* primary transcript can undergo alternative splicing at multiple sites, leading to a large number of potential splice variants that encode gene products of different lengths. More specifically, there are two alternative, mutually exclusive first exons (exon 1 and exon 2), potential exon skipping at exons 8, 15, and 17, an alternative 5' splice site in exon 24, and intron retention in the final exon 28 (*Figure 1A*). Recent studies indicate that alternative splicing can have an important impact on TRPM3 channel functionality: the presence of exon 17 is essential for inhibitory regulation of TRPM3 by the $G_{\beta\gamma}$ subunit of trimeric G proteins (*Behrendt et al., 2020*), whereas the use of alternative 5' splice sites in exon 24 leads to channel isoforms with either a short or a long pore loop, exhibiting distinct cation selectivity and sensitivity to the neurosteroid PS (*Oberwinkler et al., 2005*; *Held et al., 2022*; *Held et al., 2015*; *Held and Tóth, 2021*). However, the impact of other splicing events on channel function remains unclear, with several splice variants that can be heterologously expressed as channels with indistinguishable functional properties (*Zhao et al., 2020*; *Behrendt et al., 2020*; *Held et al., 2022*). The existence of multiple transcripts has caused ambiguity in the amino acid numbering of channel variants in patients, and the frequency of the different alternative splicing events in disease-relevant tissue is currently unknown.

To address this issue, we analyzed a publicly available RNA-seq dataset from human cerebellum and used the Sashimi plot feature of the Integrative Genomics Viewer to quantify the frequency of the different potential alternative splicing events. This analysis indicates that the large majority of transcripts in human cerebellum use exon 2 as first exon (85%), have a short pore loop (>98%), and do not undergo intron retention in exon 28 (82%). Moreover, a variable number of transcripts included the optional exons 8 (12%), 15 (41%), and 17 (85%). A construct using exon 2 as the start exon, with a short pore, including all three optional exons as well as the full exon 28 (transcript variant NM_001366147.2), corresponding to a protein of 1744 amino acids, did, however, not yield a functional channel upon heterologous expression in HEK293T cells (*Figure 1—figure supplement 1*). In contrast, a construct lacking the lowly expressed exon 8 (transcript NM_001366145.2) yielded robust TRPM3-dependent signals upon expression in HEK293T cells, as assessed using Fura-2-based calcium microfluorimetry and whole-cell patch-clamp experiments. This included robust responses to PS stimulation and potentiation by co-application of PS + clotrimazole (*Vriens et al., 2014*; *Figure 1—figure supplement 1*). We propose to use the latter transcript as the reference for variant numbering, as it represents the longest functional splice variant that includes all exons that are frequently used in human brain tissue and covers all human disease-associated *TRPM3* variants. For instance, according to this nomenclature, the recurrent variant initially indicated as p.Val837Met (*de Sainte Agathe et al., 2020*; *Dyment et al., 2019*; *Gauthier et al., 2021*; *Lines et al., 2022*) will be named p.Val1002Met.

### Identification of *TRPM3* variants in patients with neurodevelopmental disorders

Through collaborations and research networks, we ascertained 10 patients carrying *TRPM3* variants (8 females, 2 males; age 21 months to 45 years). Eight individuals carried a de novo variant, while one male patient had inherited the variant from his mildly affected father. Like patients harboring the recurrent p.Val1002Met *TRPM3* variant, patients with the novel variants presented with a neurodevelopmental disorder of variable severity, variably associated with skeletal abnormalities and epilepsy. Clinical summaries of the patients are presented in *Table 1*, summarizing the core phenotypic features of the 10 patients. The first symptoms were observed within the first year of life in 8 out of 10 patients and included hypotonia, poor visual contact or pursuit, and motor delay. In one patient, the first concern was an unstable, ataxic standing at 14 months. One patient showed a slight motor delay in childhood and had mild intellectual deficiency but was only diagnosed when his son was investigated for the same condition. Gross motor milestones were delayed in 9 out of 10 patients, autonomous

**Table 1.** Overview patient information.

| Patient | 1 | 2 | 3 | 4 | 5 (Son of 6) | 6 (Father of 5) | 7 | 8 | 9 | 10 |
|---|---|---|---|---|---|---|---|---|---|---|
| Method | Exome trio | Exome trio | Panel NGS | Panel NGS | Exome trio | Genome trio | Panel NGS | Exome trio | Exome trio | Exome trio |
| Age at last examination | 13 years | 7.5 years | 20 years | 30 months | 21 years | 45 years | 16 years | 4 years | 10 years | 3 years |
| Sex | F | F | F | F | M | M | F | F | F | F |
| OFC (SD) | 0 | −2.5 | −2 | −2.5 | +0.5 | | −1 | −1.8 | +1 | +1.5 |
| Height (SD) | 0 | na | −3 | −2.5 | +0.5 | +0.5 | −1.8 | 96.5 cm | na | 0 |
| Weight (SD) | +0.5 | na | na | −2.5 | <+5 | 0 | na | 16 kg | −0.5 | +1.8 |
| Pregnancy or delivery event | No | Placenta accreta | No | Club foot | Mild pre-eclampsia | Reduced fetal movements | No | C-section Breech position | Pre-eclampsia C-section | No |
| Maternal treatment | No | No | No | No | Enoxaparin injections | Codeine –tooth abscess | Antiretroviral therapy | No | No | Heparine therapy |
| Birth (weeks) | 40 | Full term | 41 | 40 | Full term | 42 | na | 39 | 37 | 40 |
| Birth OFC (cm) | 33.5 | na | 36 | 34.5 | na | na | 34 | 35.6 | na | na |
| Birth weight (g) | 3020 | 3500 | 3140 | 3150 | 3090 | 3360 | 3080 | 3570 | 2637 | 3430 |
| Birth length (cm) | 44 | na | 48 | 47 | na | na | 47.5 | 50.8 | na | 50 |
| First signs (age) | Unstable gait (14 months) | Poor visual contact (1 month) | Hypotonia and poor visual contact (first weeks) | Lack of visual pursuit (3 months) | Motor delay (first months) | Mild learning difficulties | Poor visual contact (first weeks) and abnormal ocular movements | Feeding difficulties hypotonia and no visual tracking (3 months) | Neonatal hypotonia and abnormal ocular movements | Motor delay (first months) |
| Hypotonia first months | No | Yes | Yes | Yes | Yes | Yes | Yes | Yes | Yes | No |
| Achieved psychomotor milestones | Able to walk unaided (ataxic) | Able to walk with aid | Able to sit, hypotonia, moves on the buttocks, and poor visual contact | Unstable head, hypotonia, and unable to follow | Able to walk unaided after motor delay | Able to sit unaided with delay | Able to walk unaided after motor delay | Unable to walk | Unable to walk | Able to sit: 12 months |
| Walking age | 25 m (ataxic) | 4 years with aid | Not acquired | Not acquired | 19 months | Normal | 20 months | Not acquired | Not acquired | 24 months |
| Ataxia | Yes | Yes | Severe hypotonia | na | No | No | Yes, improving; at 16: very mild | na (unable to walk) | No | Yes |
| Tremor | Yes | No | No | No | No | No | Yes (hands) | No | No | No |
| Dysmetria | Yes | na | na | na | No | No | Very mild and adiadococinesia | Yes | No | na |
| Dysarthria | Yes | na | na | na | No | No | na | Non-verbal | No | na |
| Dystonia | No | na | Yes | No | No | No | hand 'crispation' | No | na | No |
| Abnormal movements | Myoclonies | na | Saccadic gesticulation | No | No | No | Syncinesia | Stereotyped hand movement | na | Ataxia |
| Amyotrophy | Yes | na | na | na | na | na | na | na | na | No |
| Epilepsy (age and treatment) | No | Febrile seizures (5 years, no treatment) | No | Doubtful seizures and abundant interictal discharges (4 years, primidone) | Nocturnal epilepsy generalized tonic-clonic (7 years, no treatment) | No | No | Doubtful | Neonatal episodes (uncertain) | Yes (30 m, primidone) |

*Table 1 continued on next page*

*Table 1 continued*

| Patient | 1 | 2 | 3 | 4 | 5 (Son of 6) | 6 (Father of 5) | 7 | 8 | 9 | 10 |
|---|---|---|---|---|---|---|---|---|---|---|
| EEG: age/findings/ (Wake W/Sleep S) | na | 5 years: Background slowing (W); episodes of sharp waves in the fronto-central regions (S) | 1/8 years: Normal (W/S) | 30 months: Normal EEG 4 years: Background slowing, alpha central since 6 m, biparietal spikes, no epilepticus during slow-wave sleep (ESES; W/S) | 7 years: bifrontal synchronous spike and waves discharges (W) | na | na | 3 years: Generalized -background slowing, no epileptic discharges (W/S) | Several before 10 years: Normal (W/S) | 2.5 years: ESES (W/S) |
| Pain insensitivity (Y/N)? | No | Yes | na | na | Yes | Yes | na | na | No | Yes |
| Heat insensitivity | No | No | na | na | No | Yes | na | Yes | No | na |
| Language | Normal | Monosyllabic | Absent | Absent | Delayed | Normal | Slightly delayed then normal | Non-verbal (picture cards) | Non-verbal (picture cards) | Delayed |
| Intellectual deficiency | Normal low/ mild ID | Severe-moderate | Severe | Probably severe | Moderate | Mild | Very mild-low normal | Moderate | Severe | Yes |
| Behavior anomalies | No | No | No | No | Food-seeking | No | No | Yes | Occasional outbursts and stereotypies ‡ | Aggressivity |
| Autism spectrum disorder (Y/N)? | No | No | Poor contact (severe ID) | Poor contact (severe ID) | No | No | No | No | Yes | Yes |
| School | Special school (attention deficit and slow) | Specialized institution | Institution for children with profound intellectual and multiple disabilities | na | Special education | Normal then special education | Mainstream school with support measures, able to read, writing difficulties, and slow | Foundation for Blind School making slow progress | na | na |
| Evolution | Progress | Progress | Stable | Stable | Progress | Progress | Progress | Progress but episodes of mild psychomotor regression concomitant with behavioral fluctuations | na | Progress |
| Ocular anomalies | Strabismus, saccadic breakdown of smooth pursuit | Abnormal eye pursuit | Strabismus | No | Hypermetropia and left convergent squint | No | Nystagmus | Cortical visual impairment, nystagmus, and strabismus | Disconjugate nystagmus | Strabismus |
| Skeletal anomalies | 12th hypoplastic rib pair | Pes calcaneovalgus | Congenital hip luxation – later paralytic kyphoscoliosis | na | Left Perthes' disease | No | Valgus foot and patellar dislocation | No | Hip dysplasia | Brachydactyly |
| Others | | | | | Small genitalia, delayed puberty, and gynecomastia | Skin tags and dry palmar skin | Prognathism | Failure to thrive (milk protein allergy and GERD) | Feedings difficulties G-tube fed | Immune thrombocytopenia and hypochromic microcytic anemia |
| MRI (age) | 3 years 8 months: cerebellar atrophy, increased at 10 years | 6 months: normal 2 years 2 months: cerebellar vermis and hemispheres atrophy | 4 months: 'normal' 8 months: brainstem and cerebellar atrophy, short corpus callosum | 1 years 4 months: brainstem and cerebellar atrophy | Normal | Not done† | Very mild localized atrophy of cerebellar hemispheres | Normal | Not done* | Normal (3 years) |
| Mutation NM_001366145.2 | c.1841A>T p.(Asp614Val) | c.2305C>G p.(Leu769Val) | c.3004G>T p.(Val1002Leu) | c.3005T>G p.(Val1002Gly) | c.3019G>A p.(Gly1007Ser) | c.3019G>A p.(Gly1007Ser) | c.3019G>A p.(Gly1007Ser) | c.3376A>G p.(Asn1126Asp) | c.3376A>G p.(Asn1126Asp) | c.3397T>C p.(Ser1133Pro) |
| Inheritance | de novo | de novo | de novo | de novo | Inherited from the father | de novo | de novo | de novo | de novo | de novo |

*Table 1 continued on next page*

*Table 1 continued*

| Patient | 1 | 2 | 3 | 4 | 5 (Son of 6) | 6 (Father of 5) | 7 | 8 | 9 | 10 |
|---------|---|---|---|---|--------------|-----------------|---|---|---|----|

Abbreviations: NGS = Next Generation Sequencing; OFC = occipitofrontal circumference.

*Cerebral computed tomography (CT) at 5 years: Periventricular white matter loss.

†CT normal.

‡repetitive hyperventilation.

walking was achieved late, between 19 and 25 months in 4 out of 10 patients, walking with aid in one patient at 4 years, and walking was not achieved in 4 out of 10 patients (ages 21 months, 3 years, 10 years, and 20 years at time of study). Patients were stable or made developmental progress, but one individual had several episodes of behavioral changes with irritability, which were associated with transient degradation in motor skills. Four patients had cerebellar ataxia, and two other patients showed severe hypotonia without weakness and with nystagmus that may be related to cerebellar involvement. Language development was normal in 3 out of 10 patients, delayed in 3 patients, and absent in 4 patients. Two of the latter are able to communicate using non-verbal tools like pictograms. Intellectual deficiency was observed in all subjects, ranging from a low normal-mildly reduced IQ in 3 out of 10 patients to severe intellectual deficiency in 4 patients. There were no striking behavioral anomalies, except for patient n° 5 who displayed food-seeking behavior responsible for his obesity. One patient had febrile seizures. Epilepsy was diagnosed in only two patients: one patient had nocturnal generalized tonic-clonic seizures since 7 years of age, and a further patient had electrical status epilepticus during slow-wave sleep (ESES). Moreover, in a third patient, epilepsy was doubtful, and an epileptic therapy was started (patient n° 4). Note that sleep EEGs were not performed in all patients. None was refractory to antiseizure medication. None of the patients had hearing loss, and there was not a clearly recognizable facial dysmorphism. Notably, 7 out of 10 patients showed skeletal anomalies: hip subluxation (*Wagner et al., 2008*), patellar dislocation (*Vriens et al., 2011*), Perthes' disease (*Vriens et al., 2011*), brachydactyly (*Vriens et al., 2011*), valgus foot (*Wagner et al., 2008*), and rib hypoplasia (*Vriens et al., 2011*). Two patients had a statural growth restriction, and 3 out of 10 patients showed mild proportional secondary microcephaly. Cranial MRI was normal in 3 out of 8 patients. However, a substantial number of patients showed cerebellar atrophy (5 out of 8 patients): vermian and cerebellar hemispheres atrophy in patients n° 1–4 (*Figure 2* and *Table 1*) and a localized partial atrophy of both cerebellar hemispheres in patient n° 7 (*Figure 2—figure supplement 1*). Serial cranial MRI performed in three patients showed that the atrophy was progressive and was not present in MRI performed in the first months of life (*Figure 2*). Finally, 4 out of 10 patients exhibit pain insensitivity, and 2 patients showed heat insensitivity.

Genetic analyses identified seven novel *TRPM3* heterozygous variants in a total of 10 affected patients (*Table 1* and *Figure 1*). All variants were absent from the gnomAD database and predicted to be pathogenic according to at least two out of four prediction meta-analysis programs like REVEL (CADD, DANN, and PROVEAN; Table 3; *Quang et al., 2015*). Eight variants occurred de novo, and one was inherited from the affected father. Sequence alignment at positions D614, L769, V1002, G1007, P1102, N1126, and S1133 shows that the variants are located in highly conserved areas, both across orthologs from multiple species (*Drosophila*, zebrafish, mouse, rat, and macaca) and in the most closely related homologs within the TRPM subfamily, namely TRPM1, TRPM6, and TRPM7 (*Figure 1—figure supplement 3*). The previously identified and novel disease-associated variants localize to different regions of TRPM3, including the cytosolic N-terminus (D614V and L769V), the transmembrane region (V1002M, V1002G, V1002L, G1007S, and P1102Q), and the cytosolic C-terminus of the channel (N1126D, S1133P, and S1392T; *Figure 1—figure supplement 3*).

When mapped on the recent cryo-EM structure of TRPM3 in the closed state and in the presence of $G_{\beta\gamma}$ (pdb:8DDQ; *Zhao and MacKinnon, 2023*), it can be noted that many of the disease-associated variants cluster at the interface between the transmembrane domain and the cytosol (*Figure 1—figure supplement 2*). Notably, the N-terminal L769 sits in close proximity to V1002 and G1007 at the cytosolic end of transmembrane helix S4 and the S4-S5 linker, whereas N1126 and S1133 are located in the cytosolic TRP helix, which runs parallel to the plasma membrane, in close proximity to the S4-S5 linker. Thus, disease-associated variants affecting these residues are localized in a region that is critically involved in the gating of TRP and related voltage-gated cation channels. Interestingly, residue D614 is located in a cytosolic loop that is disordered in the cryo-EM structures. In the structure of the TRPM3 channel subunit in contact with nanobody-tethered $G_{\beta\gamma}$ (pdb:8DDQ), D614 is located just adjacent to the TRPM3-$G_{\beta\gamma}$ interaction site (*Behrendt et al., 2020*; *Zhao and MacKinnon, 2023*) raising the possibility that charge neutralization in the D614V variant may affect $G_{\beta\gamma}$-dependent channel regulation (*Figure 1—figure supplement 2*).

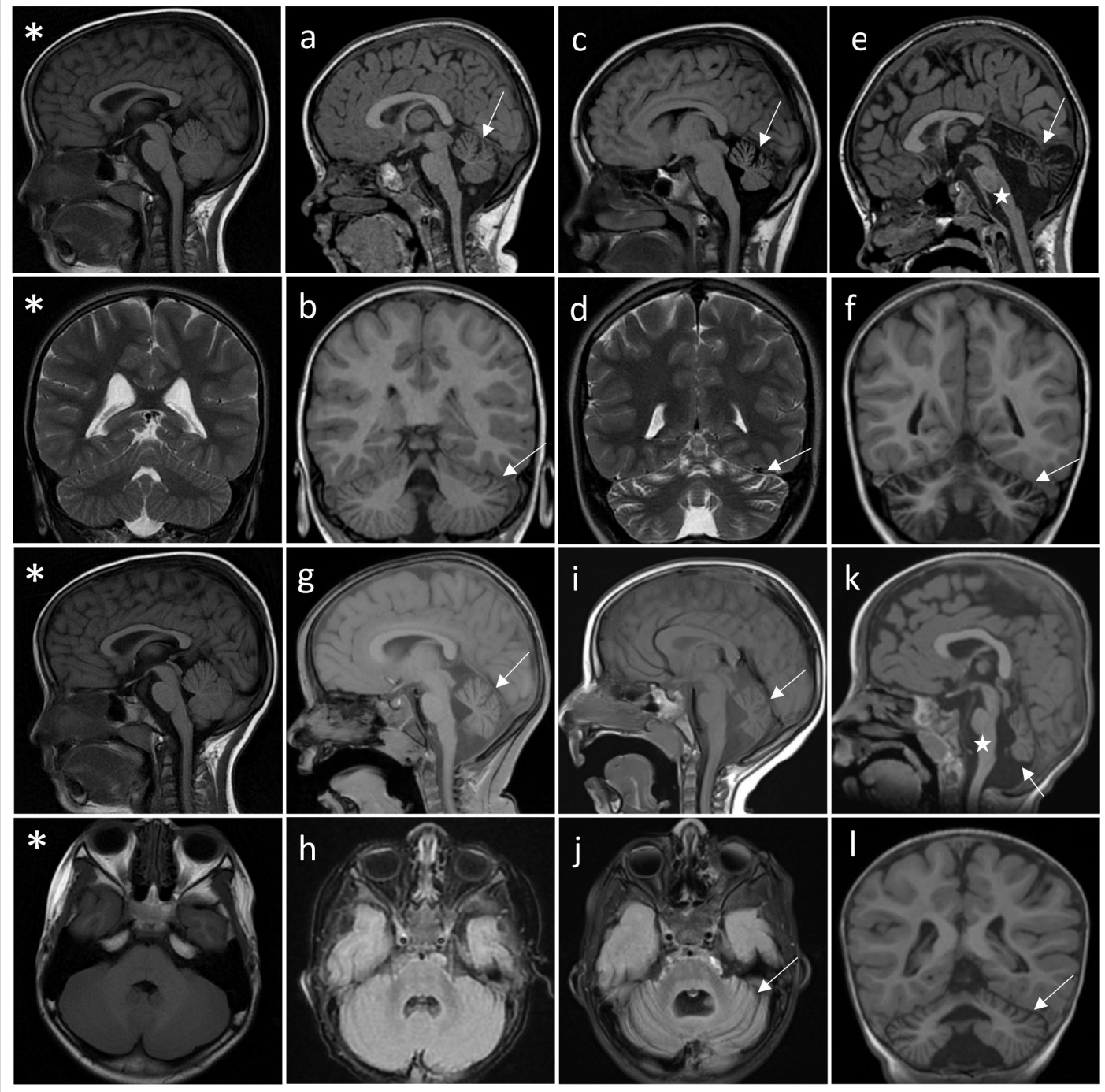

**Figure 2.** Successive MRI of several patients carrying different *TRPM3* variants. *Normal MRI: the fissures of the vermis and cerebellar hemispheres are nearly virtual. (**a–i**) MRI of the patients showing variable widening of the cerebellar fissures (arrows) reflecting cerebellar (vermis and/or hemispheres) atrophy. (**a–d**) Patient 1, MRI at 3 years 8 months showing slight atrophy of the vermis (a-sagittal T1) and cerebellar hemispheres (B-coronal T1); and majoration of the atrophy at 10 years (c-sagittal T1 and d-coronal T2), (**e–f**) Patient 3; MRI at 8 years 6 months: severe atrophy of the vermis (arrow) and brainstem (star), and atrophy of the cerebellar hemispheres (sagittal and coronal T1). (**g–j**) successive MRIs in patient showing progressive atrophy (**g**: 2 years 2 months; **h**: 6 months; **i–j**: 4 years 2 months). (**k and l**) Patient 4; MRI at 1 years 4 months: small vermis, thin brainstem (star) and atrophy of the cerebellar hemispheres (sagittal and coronal T1).

The online version of this article includes the following figure supplement(s) for figure 2:

**Figure supplement 1.** MRI of patient 7 and patient 8.

## Functional expression of disease-related *TRPM3* variants

To address whether the seven newly identified variants affect TRPM3 channel activity, we introduced the corresponding point mutations into a previously well-characterized human TRPM3 expression vector for heterologous expression (*Van Hoeymissen et al., 2020*). Fura-2-based calcium imaging in transfected HEK293T cells was used to evaluate basal channel activity and responses to the TRPM3 antagonist primidone (*Krügel et al., 2017*) and to investigate the sensitivity toward stimulation via the

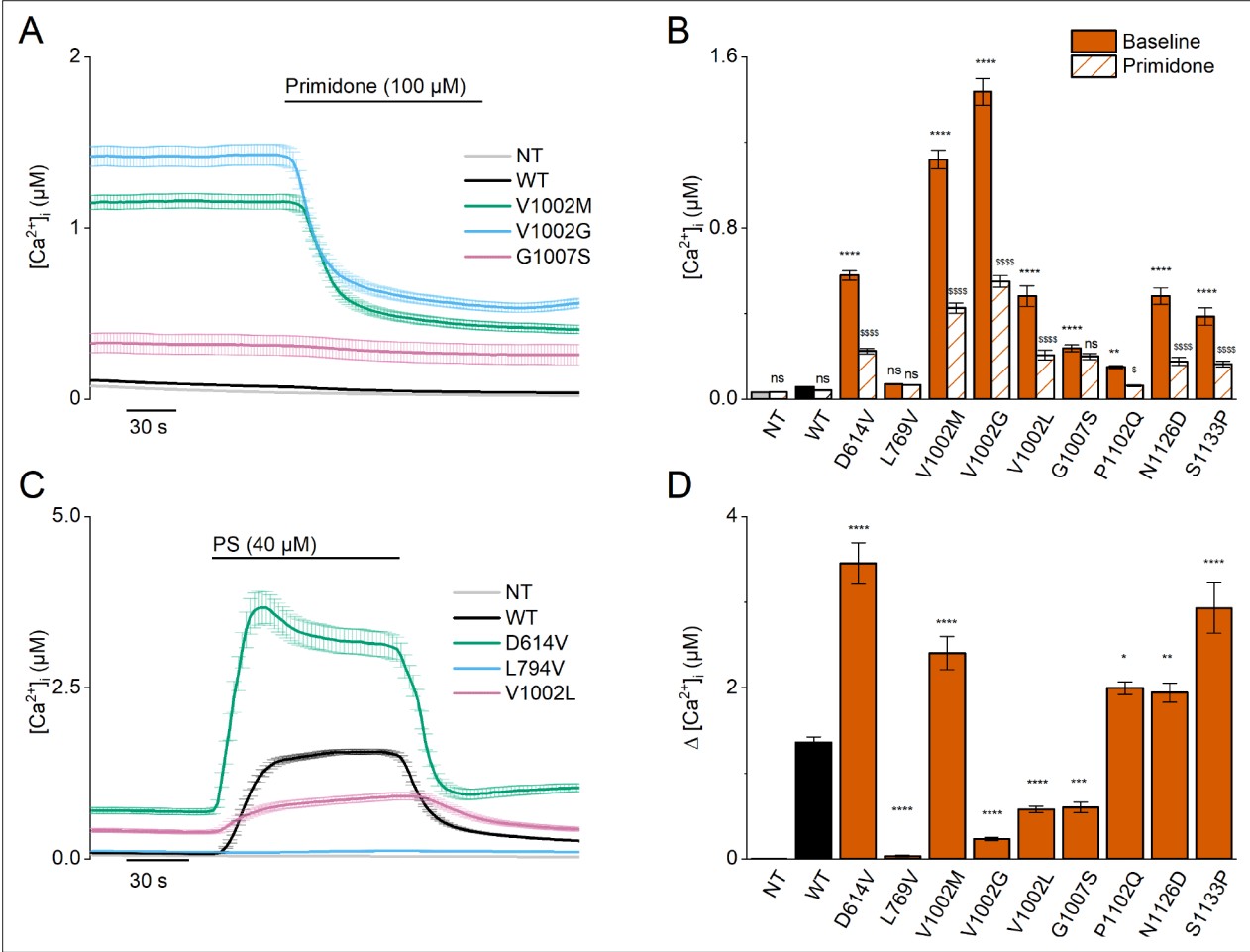

**Figure 3.** Homozygous mutant expression in HEK293T cells. (**A**) Time course of intracellular calcium concentrations ([Ca$^{2+}$]$_i$) (± SEM) upon application of the TRPM3 inhibitor primidone (100 µM) for wild-type (WT; black; n=449), and homozygous V1002M (green; n=271), V1002G (blue; n=257), and G1007S (red; n=409) transfected HEK293T cells, and non-transfected (NT; gray; n=982; N=3 independent measurements). (**B**) Mean basal intracellular calcium concentrations, [Ca$^{2+}$]$_i$, in the absence (full bars) and presence of primidone (100 µM; striped bars). Data are represented as mean ± SEM, using a two-way ANOVA with Sidak's posthoc test. p-Values of baseline vs WT: NT (p=0.8376), D614V (p<0.0001), L794V (p>0.9999), V1002M (p<0.0001), V1002G (p<0.0001), V1002L (p<0.0001), G1007S (p<0.0001), P1102Q (p=0.0018), N1126D (p<0.0001), S1133P (p<0.0001); p-values baseline vs primidone: NT (p>0.9999), WT (p=0.9994), D614V (p<0.0001), L794V (p>0.9999), V1002M (p<0.0001), V1002G (p<0.0001), V1002L (p<0.0001), G1007S (p=0.5663), P1102Q (p=0.0138), N1126D (p<0.0001), S1133P (p<0.0001). (**C**) Time course of [Ca$^{2+}$]$_i$ (± SEM) for WT (black) (n=243), D614V (green) (n=220), L794V (blue) (n=420) and V1002L (red) (n=264) transfected HEK293T cells, and non-transfected (NT; gray) (n=452) upon application of pregnenolone sulfate (PS; 40 µM) (N=3 independent measurements). (**D**) Corresponding calcium amplitudes of the PS response, represented as mean ± SEM, using a Kruskal-Wallis ANOVA with Dunnett's posthoc test (p-values vs WT: D614V (p<0.0001), L794V (p<0.0001), V1002M (p<0.0001), V1002G (p<0.0001), V1002L (p<0.0001), G1007S (p=0.0006), P1102Q (p=0.0102), N1126D (p=0.0098), S1133P (p<0.0001)). For these experiments, the isoform corresponding to GenBank AJ505026.1 was used.

The online version of this article includes the following figure supplement(s) for figure 3:

**Figure supplement 1.** Elevated basal activity in HEK293T cells expressing TRPM3 variants.

**Figure supplement 2.** PS-induced calcium influxes in TRPM3-developmental and epileptic encephalopathy (DEE) mutants.

**Figure supplement 3.** Pregnenolone Sulfate dose dependency of the TRPM3 variant G1007S.

**Figure supplement 4.** Fluorescence Intensity of channel-linked yellow fluorescence protein (YFP) in cells expressing WT TRPM3 and TRPM3 variants.

agonist PS. In line with earlier work, cells transfected with the WT TRPM3 construct showed a small increase in basal intracellular calcium concentration ($[Ca^{2+}]_i$) compared to non-transfected (NT) cells. In WT transfected cells, the cytosolic $[Ca^{2+}]_i$ decreased slightly in response to treatment with a high dose of primidone (100 μM; *Figure 3A and B*, *Figure 3—figure supplement 1* and *Figure 4—figure supplement 1*). The primidone dose was based on earlier work, demonstrating a lower sensitivity of *TRPM3* variants p.Val1002Met and p.Pro1102Gln toward primidone stimulation (*Van Hoeymissen et al., 2020*; *Zhao et al., 2020*).

When overexpressing the newly identified TRPM3 variants, with the exception of the L769V variant, we consistently observed basal $[Ca^{2+}]_i$ levels that were significantly higher compared to NT cells or cells transfected with WT-TRPM3, an effect that was most pronounced for the V1002G variant (*Figure 3A and B*, *Figure 3—figure supplement 1* and *Figure 4—figure supplement 1*). Except for the L769V and the G1007S variant, application of primidone caused a reduction of $[Ca^{2+}]_i$, albeit not to the levels of NT cells (*Figure 3A and B*, *Figure 3—figure supplement 1* and *Figure 4—figure supplement 1*). These results indicate that the patient variants lead to increased basal TRPM3 activity. A more mixed result was obtained when assessing the responses of the variants to the neurosteroid agonist PS. Compared to WT-TRPM3, PS responses were increased for the D614V, N1126D, and S1133P variants and reduced for the L769V, V1002G, V1002L, and G1007S variants (*Figure 3C and D*, *Figure 3—figure supplement 2* and *Figure 4—figure supplement 2*).

Note that all described patients are heterozygous for the specific TRPM3 variants and thus possess one WT allele. To mimic the patient situation in our cellular assay, we performed experiments in HEK293T cells transfected with a mixture of cDNA encoding WT and variant TRPM3 in a 1:1 ratio. Under these conditions, significantly higher basal $[Ca^{2+}]_i$ levels for all WT/variant mixtures were observed compared to cells expressing only WT-TRPM3, with the exception of the P1102Q and N1126D variant, where the elevated basal $[Ca^{2+}]_i$ levels were found not to be significant. Moreover, under these heterozygous conditions, primidone caused a reduction in basal $[Ca^{2+}]_i$ for all tested variants with higher basal $[Ca^{2+}]_i$ levels (*Figure 4A and B*, *Figure 3—figure supplement 1* and *Figure 4—figure supplement 1*). Finally, cells co-expressing WT-TRPM3 with any of the newly discovered disease-associated variants consistently exhibited larger PS-induced $Ca^{2+}$ responses (*Figure 4C and D*, *Figure 3—figure supplement 2* and *Figure 4—figure supplement 2*).

Since the calcium imaging experiments suggested that the L769V and G1007S variants had opposite effects on channel activity when expressed alone vs combined with WT subunits, we further evaluated their functionality using whole-cell patch-clamp electrophysiology. We measured whole-cell TRPM3 currents in response to a voltage step protocol ranging from –160 mV to +160 mV (*Figure 4—figure supplement 3*), both at baseline and upon stimulation with PS. At room temperature, and in the absence of activating ligand, WT-TRPM3 carries a small, outwardly rectifying current (*Vriens et al., 2011*; *Wagner et al., 2008*). We observed that cells expressing WT + L769 V, G1007S, and WT + G1007 S produced robust outwardly rectifying currents, whereas currents in cells expressing only L769V were not different from those in NT cells (*Figure 4—figure supplement 3B*). Upon stimulation with PS, robust outwardly rectifying currents were evoked in cells expressing WT, WT + L769 V, G1007S, and WT + G1007 S, whereas currents in cells expressing L769V were not different from those in NT controls (*Figure 4—figure supplement 3D*). Notably, in cells expressing WT + L769V, we measured an increase in inward current at –160 mV when compared to NT controls, reminiscent of the inwardly rectifying current component observed in the V1002M variant (*Figure 4—figure supplement 3E*; *Van Hoeymissen et al., 2020*).

Since the G1007S variant showed a reduced $Ca^{2+}$ response to PS stimulation in the homozygous condition, but a significantly increased PS response in the heterozygous condition, we tested the concentration dependence of the response to PS for this variant using a calcium-based assay performed in a plate-reader system. These experiments revealed a shift of the concentration-response curve to lower PS concentrations for the heterozygous, but not for the homozygous condition ($EC_{50}$ value of 2.9±0.3 μM, 4.0±0.5 μM, and 1.1±0.1 μM for, respectively, WT-TRPM3, homozygous G1007S, and heterozygous WT + G1007 S transfected HEK293 cells; *Figure 3—figure supplement 3*). Taken together, these data demonstrate that all patient variants are dominant gain-of-function mutations, provoking significantly increased basal activity resulting in cellular calcium overload, as well as enhanced PS-induced responses.

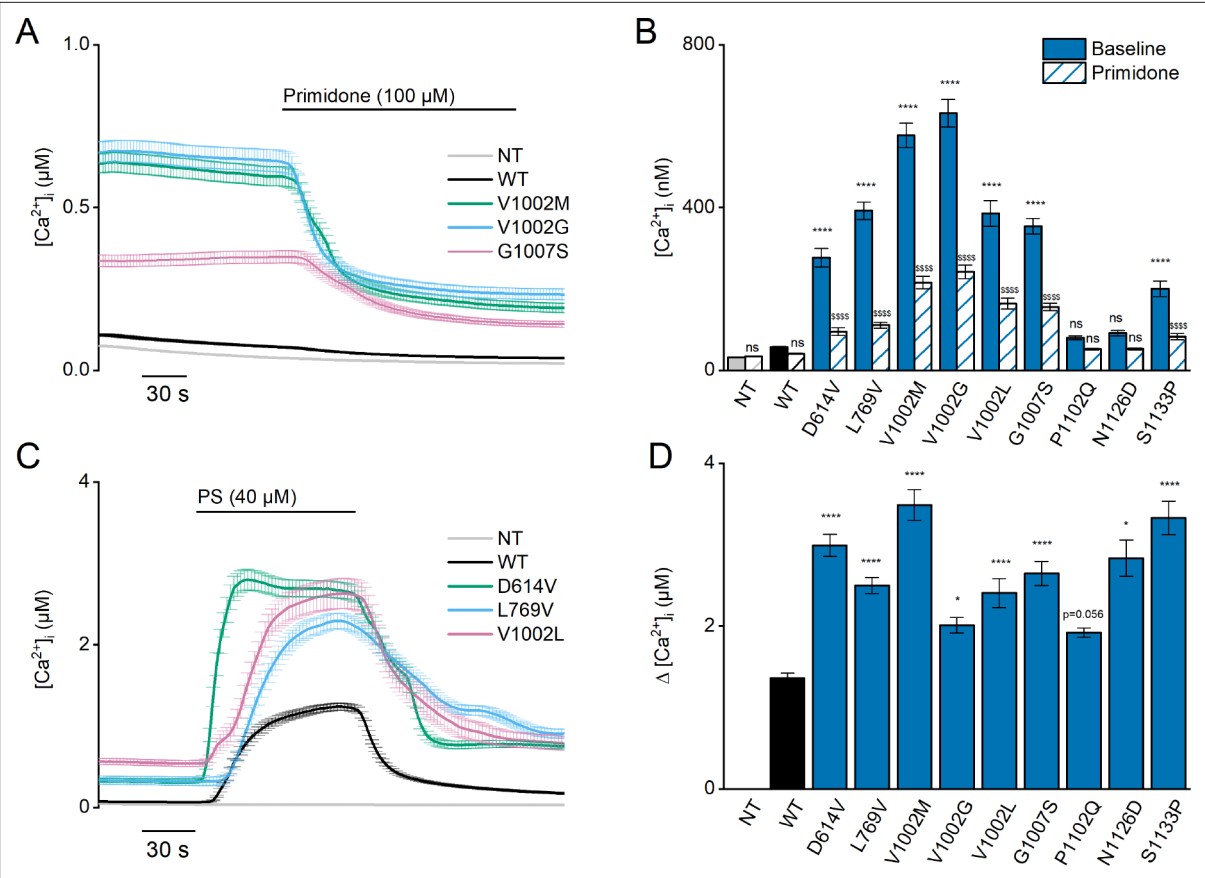

**Figure 4.** Heterozygous mutant + wild-type (WT) expression in HEK293T cells. (**A**) Time course of $[Ca^{2+}]_i$ ± SEM upon application of the TRPM3 inhibitor primidone (100 µM) for WT (black; n=449), and heterozygous WT + V1002 M (green; n=373), WT + V1002 G (blue; n=482), and WT + G1007 S (red) (n=561) transfected HEK293T cells, and non-transfected (NT; gray; n=982; N=3 independent measurements). (**B**) Mean basal $[Ca^{2+}]_i$ ± SEM in the absence (full bars) and presence of primidone (striped bars). A two-way ANOVA with Sidak's posthoc test was used. p-Values of baseline vs WT: NT (p=0.6733), D614V (p<0.0001), L794V (p<0.0001), V1002M (p<0.0001), V1002G (p<0.0001), V1002L (p<0.0001), G1007S (p<0.0001), P1102Q (p=0.9249), N1126D (p=0.5539), and S1133P (p<0.0001); p-Values baseline vs primidone: NT (p>0.9999), WT (p=0.9994), D614V (p<0.0001), L794V (p<0.0001), V1002M (p<0.0001), V1002G (p<0.0001), V1002L (p<0.0001), G1007S (p<0.0001), P1102Q (p=0.8205), N1126D (p=0.5132), and S1133P (p<0.0001). (**C**) Time course of $[Ca^{2+}]_i$ (± SEM) for WT (black; n=243), WT + D614V (green; n=281), WT + L794V (blue; n=497) and WT + V1002L (red; n=276) transfected HEK293T cells, and non-transfected (NT; gray; n=452) upon application of pregnenolone sulfate (PS; 40 µM; N=3 independent measurements). (**D**) Corresponding calcium amplitudes of the PS response, represented as mean ± SEM, using a Kruskal-Wallis ANOVA with Dunnett's posthoc test (p-values vs WT: D614V [p<0.0001], L794V [p<0.0001], V1002M [p<0.0001], V1002G [p=0.0155], V1002L [p<0.0001], G1007S [p<0.0001], P1102Q [p=0.0564], N1126D [p=0.0277], and S1133P [p<0.0001]). For these experiments, the isoform corresponding to GenBank AJ505026.1 was used.

The online version of this article includes the following figure supplement(s) for figure 4:

**Figure supplement 1.** Individual data points of intracellular calcium concentrations at baseline and upon application of the TRPM3 inhibitor primidone.

**Figure supplement 2.** Individual data points of intracellular calcium amplitudes upon application of the TRPM3 agonist pregnenolone sulfate (PS).

**Figure supplement 3.** Baseline and pregnenolone sulfate (PS)-induced current densities of the L769V and G1007S substitution.

To investigate whether alterations in the cellular expression levels of the variant TRPM3 channel subunits contribute to the increased channel activity, we measured single-cell yellow fluorescence protein (YFP) fluorescence and basal $[Ca^{2+}]_i$ levels in HEK293T cells transfected with the different TRPM3 variants (either alone or in combination with WT-TRPM3) and compared them with levels in cells transfected with WT-TRPM3 on the same experimental day. For variants V1002L, G1007S, N1126D, and S1133P, YFP levels were not higher than in cells expressing WT-TRPM3, similar to our earlier observations for the V1002M and P1102Q variants (*Van Hoeymissen et al., 2020*; *Figure 3—figure supplement 4*). In cells expressing the L769V variant in the absence of WT subunits, which do not show a response to the channel antagonist primidone or the channel agonist PS (*Figure 3*), cellular YFP values were significantly reduced compared to WT (*Figure 3—figure supplement 4*). However,

normal YFP levels were found in cells co-expressing the L769V variant with WT (*Figure 3—figure supplement 4*). Finally, in the case of D614V and V1002G variants, cellular YFP levels were significantly higher than WT control, both under homozygous and heterozygous conditions (*Figure 3—figure supplement 4*). To evaluate whether the increased basal activity for these variants is a mere consequence of the increased channel expression levels, we plotted basal $[Ca^{2+}]_i$ vs cellular YFP levels for individual cells expressing either WT, variant, or WT + variant. This analysis revealed that, in cells with similar YFP fluorescence, basal $[Ca^{2+}]_i$ levels were consistently higher for cells expressing the D614V and V1002G variants, either alone or with WT subunits (*Figure 3—figure supplement 4*).

Taken together, these data indicate that elevated basal $[Ca^{2+}]_i$ levels in cells expressing disease-associated TRPM3 variants can be attributed to increased activity of individual channels rather than to increased channel protein expression.

## Discussion

TRPM3 is a calcium-permeable cation channel belonging to the melastatin subfamily of TRP channels. It functions as a heat- and neurosteroid-activated channel in peripheral sensory neurons of the trigeminal and dorsal root ganglia (*Vangeel et al., 2020*) but is also expressed in other tissues, including the CNS (*Figure 5—figure supplement 1*). Recently, three de novo variants in the *TRPM3* gene were identified in patients with DEE (*de Sainte Agathe et al., 2020*; *Dyment et al., 2019*; *Gauthier et al., 2021*; *Kang et al., 2021*; *Lines et al., 2022*). The 16 patients heterozygous for the common recurrent variant (V1002M), share a number of clinical features including GDD, moderate to severe ID, with or without childhood-onset epilepsy (*Lines et al., 2022*).

Here, we describe patients carrying one of seven novel de novo variants in the *TRPM3* gene. These patients present with a large spectrum of neurodevelopmental symptoms, including GDD and ID, variably associated with seizures, skeletal anomalies, and insensitivity to pain and heat. Intellectual deficiency as well as motor disabilities are highly variable, from polyhandicap to very mild impact compatible with parentality and normal life in adulthood. We highlight a cerebellar phenotype (ataxia or severe hypotonia, nystagmus or abnormal oculomotricity, and cerebellar atrophy) in more than half of the patients as a novel feature of the TRPM3-linked disease pattern. The cerebellum is known to integrate neuronal networks coupling motor function with cognition, emotional skills, and language (*Leto et al., 2016*; *Sathyanesan et al., 2019*; *Schmahmann, 2019*). Distortion of cerebellar development could induce (cerebellar) diseases (*Sathyanesan et al., 2019*). Abundant RNA expression of TRPM3 in the brain is already described (*Held and Tóth, 2021*); however, limited knowledge is available on the functional expression of TRPM3 in the CNS, including cerebellar Purkinje cells and oligodendrocytes (*Zamudio-Bulcock et al., 2011*; *Held and Tóth, 2021*; *Hoffmann et al., 2010*; *Oberwinkler and Philipp, 2014*). By reanalyzing publicly available single cell RNA sequencing datasets of the developing (*Aldinger et al., 2021*) and adult (*Lake et al., 2018*) cerebellum and adult cortex (*Velmeshev et al., 2019*), it becomes apparent that TRPM3 is robustly expressed in different cell types of these brain regions (*Figure 5* and *Figure 5—figure supplement 2*). Highest expression of TRPM3 in the adult cortex (*Velmeshev et al., 2019*) was found in layer 5/6 neurons (*Figure 5—figure supplement 2A, C, E*). Furthermore, TRPM3 expression was observed in distinct cellular clusters of the adult cerebellum (*Lake et al., 2018*), including neuronal (e.g. distinct cerebellar granule cells and Purkinje neurons) and non-neuronal (e.g. cerebellar-specific astrocytes) cell types (*Figure 5B, D and F*). Interestingly, prominent TRPM3 expression was observed in early stages of the developing cerebellum (*Figure 5—figure supplement 2B, D*) and detected in different cell types (*Figure 5A, C and E*), including excitatory cerebellar and unipolar brush cell interneurons, and Purkinje cells (*Aldinger et al., 2021*). Highest expression is noted in cells of the rhombic lip (*Figure 5A, C, E and G*), where TRPM3 is expressed in all four zones: choroid plexus epithelium, intermediate zone, subventricular zone, and ventricular zone (*Aldinger et al., 2021*). Interestingly, low RNA levels were detected in different muscle types (*Figure 5—figure supplement 1A*). This might suggest that hypotonia in most of the patients (*Table 1*) is most likely caused by malfunctioning of the neuronal innervation of the skeletal muscles, probably related to a cerebellar defect. Taken together, the expression data of TRPM3 in the CNS points toward an important role of the channel in specific brain regions, both in the early developmental and adult stage.

As it is known in some other congenital ataxia, we observed that cerebellar atrophy could be progressive in a patient who has a clinically non-progressive disorder. This should encourage repetition

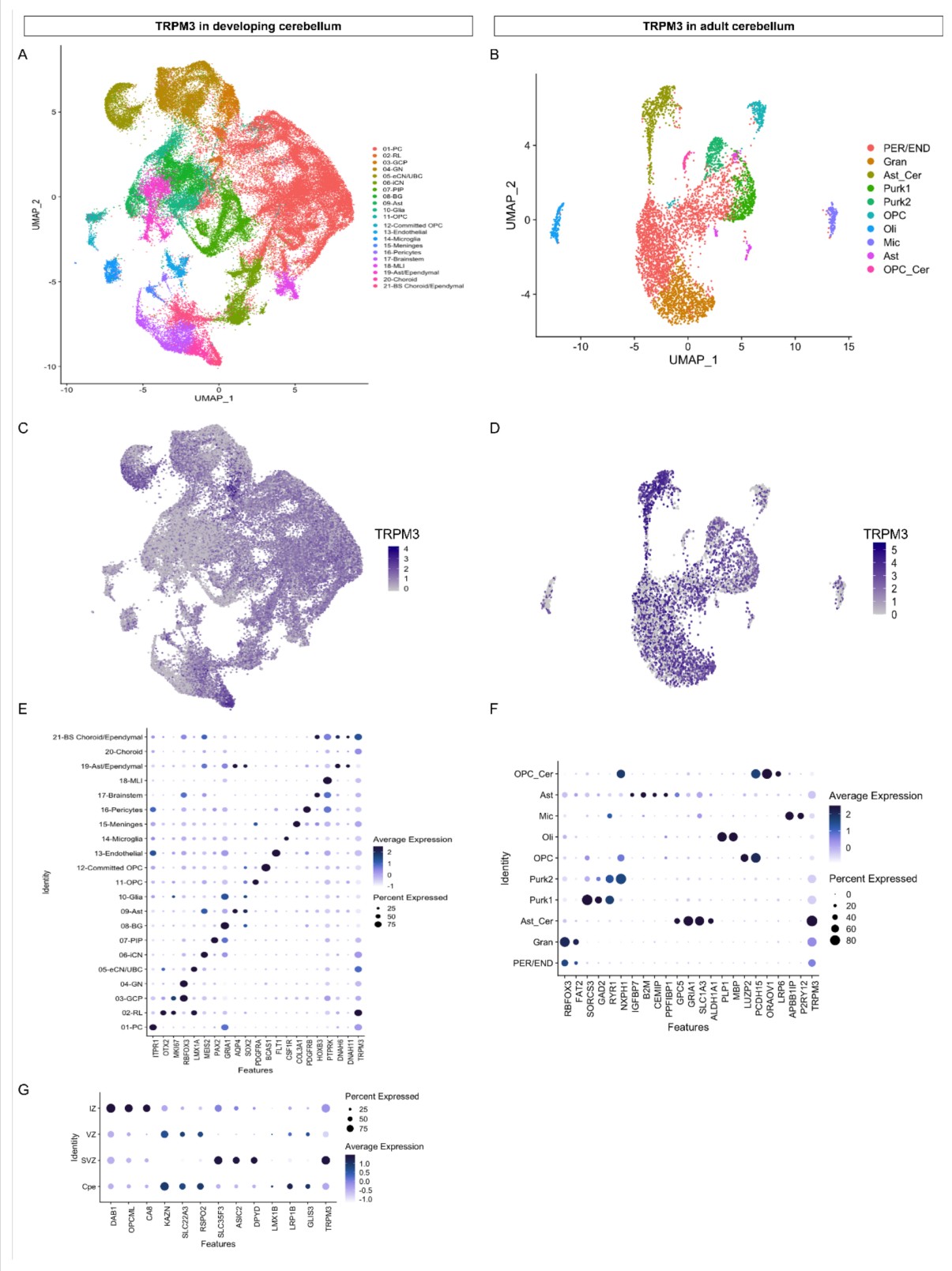

**Figure 5.** TRPM3 expression in the human cerebellum. (**A and B**) Uniform Manifold Approximation and Projection (UMAP) visualization of human cerebellar nuclei annotated on the basis of marker genes for the developing (**A**) and adult cerebellum (**B**). (**C and D**) The same UMAP visualization of human cerebellar nuclei as panel (**A**) and (**B**), now representing the expression of TRPM3 for the developing (**C**) and adult cerebellum (**D**), respectively. (**E and F**) Dot plot showing the expression of one selected marker gene per cell type for the developing (**E**) and adult cerebellum (**F**). The size of the dot

*Figure 5 continued on next page*

*Figure 5 continued*

represents the percentage of nuclei within a cell type in which that marker was detected, and its color represents the average expression level. (**G**) Dot plot showing the expression of one selected marker gene per region of the rhombic lip (RL). Data set of TRPM3 in developing cerebellum adapted from *Aldinger et al., 2021*. Data set of TRPM3 in adult cerebellum adapted from *Lake et al., 2018*.

The online version of this article includes the following figure supplement(s) for figure 5:

**Figure supplement 1.** RNA expression of TRPM3 in humans.

**Figure supplement 2.** TRPM3 expression in adult cortex and developing cerebellum.

of brain imaging in the early years of life in ataxic children. Although clinical variability, even intra-familial, is observed with the same mutation, we point on some possible genotype-phenotype correlation to be confirmed in larger series, with p.(Gly1007Ser) being associated with a milder phenotype and p.(Asn1126Asp) with a more severe phenotype.

Functional characterization of the newly identified *TRPM3* variants revealed a pronounced gain-of-function phenotype for all variants in the heterozygous (WT + mutant) condition. Typically, cells expressing these variants along with the WT channel displayed an elevated intracellular calcium concentration and increased calcium responses upon stimulation with PS. These characteristics are consistent with the previously described gain-of-function *TRPM3* variants p.(Val1002Met) and p.(Pro1102Gln) (*Van Hoeymissen et al., 2020*; *Zhao et al., 2020*).

The different disease-associated gain-of-function variants occur in different parts of the TRPM3 channel, including the cytosolic N-terminus, the transmembrane region, and the cytosolic C-terminus, suggesting that they increase channel activity via distinct mechanisms (*Figure 1—figure supplement 2*). Based on fluorescent tagging of the channels, we can exclude increased protein expression levels as an underlying mechanism, suggesting that the variants affect channel gating or membrane localization (*Figure 3—figure supplement 4*). In this respect, it is interesting to note that when mapped on the recent cryo-EM structure of TRPM3, the affected residues L769, V1002, G1007, N1126, and S1133 cluster at the interface between the transmembrane domain and the cytosol, in the gating of TRPM3 and related channels with six transmembrane domains (*Figure 1—figure supplement 2*; *Owsianik et al., 2006*). The D614V variant is located adjacent to exon 17, in a region that is not resolved in the cryo-EM structures of the entire TRPM3 channel but forms the N-terminal interaction site for the binding of the $G_{\beta\gamma}$ subunits of trimeric G proteins, leading to channel inhibition (*Badheka et al., 2017*; *Dembla et al., 2017*; *Quallo et al., 2017*). Intriguingly, our previous results showed a reduced sensitivity toward $G_{\beta\gamma}$-dependent modulation for the p.(Val1002Met) variant compared to WT transfected cells (*Van Hoeymissen et al., 2020*). Further work will be needed to clarify whether increased activation of the D614V variant occurs (partly) via disturbed binding of the $G_{\beta\gamma}$ subunits. Interestingly, there are known mutations in the upstream genetic component guanine nucleotide-binding protein beta 1 (GNB1) causing severe neurodevelopmental disability, hypotonia, and seizures (*Petrovski et al., 2016*). In addition, other mutations in GNB2 (*Fukuda et al., 2020*) and GNB5 (*Lodder et al., 2016*) are observed in patients with neurodevelopmental disorders. Potentially, these neurodevelopmental phenotypes could be partially explained via dysregulation of downstream TRPM3 activity.

Notably, L769V was the only variant that exhibited no functional activity when expressed in the absence of WT channel subunits, whereas it caused a gain-of-function in the presence of WT subunits, which corresponds to the situation in the cells of the heterozygous patients. One possible explanation could be that the variation in the N-terminus affects proper trafficking of the homotetrameric channel to the plasma membrane and that this trafficking deficit can be rescued by heteromeric channels composed of WT and variant subunits. Clearly, further research is needed to pinpoint the molecular and cellular mechanisms that lead to the gain-of-function caused by the channel variants and to reveal the pathophysiological mechanisms whereby altered channel function leads to the complex symptoms encountered by the patients.

Importantly, the increased channel activity under basal conditions and associated increased basal calcium levels observed with all the characterized disease-associated *TRPM3* variants can be blocked by application of a high dose of the antiepileptic drug primidone, which has been identified as a direct TRPM3 antagonist in in vitro studies and in animal models (*Krügel et al., 2017*). Since plasma levels in subjects taking primidone are expected to be sufficiently high to cause significant inhibition of TRPM3

activity, it will be of great interest to evaluate whether the drug can alleviate or revert symptoms in patients carrying disease-associated *TRPM3* variants.

Taken together, we have described seven novel variants in patients with a *TRPM3*-associated neurodevelopmental syndrome. The clinical phenotype of these patients is variable, with GDD and ID as consistent features. Epileptic seizures, skeletal anomalies, and pain insensitivity are observed in a subset of patients, and more than half of the patients presented with a cerebellar phenotype (ataxia or severe hypotonia, nystagmus, or abnormal oculomotricity) associated with a progressive cerebellar atrophy. We propose that *TRPM3* should be added in NGS panels designed for the diagnosis of epilepsy, ID, and congenital ataxia. The disease-associated variants consistently result in a pronounced gain of channel function, providing strong support for the hypothesis that increased channel activity, potentially leading to neuronal hyperexcitability and cellular calcium overload, underlies a spectrum of TRPM3 channelopathies.

## Materials and methods
### Patient recruitment and genomic sequencing
Patients with *TRPM3* variants were recruited through GeneMatcher (*Sobreira et al., 2015*) or previous collaboration between the participating teams. Clinical data of each patient as well as brain images were reviewed by the clinicians (geneticists, neuropediatricians, and radiologists) from the participating centers. Sequencing and genetic analyses were performed in the respective centers on a clinical basis. The study was performed in accordance with the guidelines specified by the institutional review boards (IRBs) and ethics committees at each institution. In seven patients, trio whole exome sequencing (WES) was performed (patients 1, 2, 5, 6, and 8–10), while comprehensive multigene panel (NGS targeted panel analysis) was performed in three other patients (patients 3, 4, and 7). Confirmation of the variants was performed using targeted Sanger sequencing in probands and parents. All parents agreed on sharing and publishing the patients data.

### Ethics statement
Patients 1, 3, 4, and 7: For these patients, clinical genetic services and a genetic testing were done as part of routine clinical care. Written informed consent was obtained from the parents of the probands for molecular genetic analysis and possible publication of the anonymized clinical data. The study was done in accordance with local research and ethics requirements. Patient 2: Parents signed an informed consent, received a genetic counseling before and after the analysis, and the genetic study was performed in accordance with German and French ethical requirements and laws. Data sharing was performed using anonymized genetic and clinical information. Patient 5: The patient was identified via the Deciphering Developmental Disorders (DDD) study, which was granted by the UK ethical approval by the Cambridge South Research Ethics Committee (10/H0305/83). Patient 6: This patient was identified through diagnostic testing as part of their routine clinical care within the UK National Health Service and so no specific institutional ethical approval was required. Patient 8: Informed consent for participation was obtained from subjects themselves or, where necessary, their parents. The study was completed per protocol in accordance with the Declaration of Helsinki with local approval by the Children's Hospital of Philadelphia (CHOP) IRB (15–12226). Patient 9: The participating family signed the IRB research protocol of the University of Pennsylvania division of Neurology. Patient 10: The participating family, consisting of the mother, father, and female proband, provided written consent and was enrolled into the Center for Rare Childhood Disorders (C4RCD) research protocol at the Translational Genomics Research Institute (TGen). Written consent for the proband under the age of 18 years was obtained from the parents. The study protocol and consent documents were approved by the Western IRB (WIRB # 20120789). The retrospective analysis of epilepsy patient data was approved by the local ethics committees of the Charité (approval no. EA2/084/18).

### Patient information
#### Patient 1
The proband is the first child from non-consanguineous healthy parents. She has a healthy brother. No medical history in the family was reported. She was born at term without significant pregnancy history. Birth parameters were within normal limits. Initially, psychomotor development was described

as normal, with head holding before 3 months, and she was able to sit at around 8 months. The first parental concerns occurred at 14 months as the first steps were abnormally unstable. A neurological assessment at 2 years noted ataxic of stance and gait, and dysmetria. The child evolved with a motor delay and acquired autonomous but unsteady walking at 25 months. She evolved with cerebellar motor symptoms, ataxia of stance and gait, dysmetria, adiadochokinesia, intention tremor, dysarthria, and saccadic breakdown of smooth pursuit with strabismus. The onset of speech appeared normally despite dysarthria. During its evolution, until the last examination at 13 years, she made motor progress, walking is less ataxic, but she had persistent tremor in fine motor skills and severe dysarthria. She never had seizure. She had a neuropsychological assessment at 10 years which showed mild ID (total IQ not calculable because of the heterogeneity of the scores). Difficulties were more marked in executive functions (perseveration, distractibility, and disorder of emotion regulation) and visuospatial function. She was in normal school and had intervention aids such as speech therapy, physiotherapy, and adapted educational equipment. She needed help because of lack of autonomy and emotional immaturity. Brain MRIs performed at 3 years 8 months showed cerebellar atrophy, predominant on the superior cerebellar vermis (*Figure 2*), which had increased at 10 years. Trio WES identified a de novo, heterozygous, missense variant in the *TRPM3* gene (M_001366145.2:c.1841A>T; p.[Asp614Val]).

For this patient, clinical genetic services and a genetic testing were done as part of routine clinical care. Written informed consent was obtained from the parents of the probands for molecular genetic analysis and possible publication of the anonymized clinical data. The study was done in accordance with local research and ethics requirements.

## Patient 2

She was born from healthy unrelated parents. Pregnancy, delivery, and birth parameters were normal. Poor visual contact was noticed in the first month of life. Ophthalmological examination revealed a severely reduced central visual acuity, photophobia, normal funduscopy, and a mitigated photopic and scotopic ERG responses with normal flash evoked visual responses. These findings were compatible with a mainly central retinal dysfunction. She also had a Mittendorf cataract at the right eye. MRI performed at 6 months was considered normal. At 11 months, she was able to smile, but she had axial hypotonia, poor visual contact, was unable to sit, and had hand wringing. She had no feeding difficulties. MRI was performed at 16 months and showed a slight enlargement of cerebellar sulci. She progressed slowly, and at 5 years, she was able to walk with assistance, with unsteady gait. She had febrile seizures. Wake EEG showed background slowing without epileptic discharges, and she had some episodes of sharp waves in the fronto-central regions during the night. Head circumference (HC) was at –2.5 SD. She had pes calcaneovalgus and slight dysmorphism, with wide nasal bridge, thin upper lip, and pointed chin. Parents reported a low reactivity to pain when specifically asked the question. At that time, MRI showed cerebellar atrophy (*Figure 2*), confirmed by MRI at 4 years. Trio exome sequencing identified the NM_001366145.2:c.2305C>G, p.(Leu769Val) de novo, heterozygous *TRMP3* missense variant.

Parents signed an informed consent and received a genetic counseling before, and after the analysis, and the genetic study was performed in accordance with German and French ethical requirements and laws. Data sharing was performed using anonymized genetic and clinical information.

## Patient 3

Patient 3 is the second child of healthy parents. Her brother was healthy. She was born at 41 weeks after an uneventful pregnancy. Her birth parameters were normal (weight 3140 g; length 48 cm; HC 36 cm). Left hip luxation was treated by abduction splint for 6 months. She had hypotonia and feeding difficulties from the first weeks, and her development was significantly delayed. She was able to attain head control and independent sitting but was never able to stand alone or walk. She had a paralytic thoracolumbar kyphoscoliosis requiring surgical treatment at 15 years old. She had failure to thrive, and her HC growth slowed in the first months to reach –2 SD. At 20 years of age, she had profound IDs and multiple disabilities. She was able to grasp the objects with dysmetria. She had no language due to her severe intellectual deficiency and was not able to communicate even through visual contact. Brainstem Auditorial Evoked Potential showed hearing threshold at 20 db on the left and 30 db on the right. She had stereotypies, and she never had seizures. Sleep and wake EEG were both normal (one year and 8 years). She experienced feeding difficulties with selective food, and insufficient weight

and height gain were on going. MRI performed at 4 months was normal, but the following MRI at 8 month's old showed global cerebellar atrophy that became marked at 8 years 6 months (*Figure 2*) and a short corpus callosum (−3 SD). Array CGH was normal. Analysis of our NGS targeted congenital ataxia panel including *TRPM3* (panel designed after the diagnosis made in patient 1 using WES), identified the NM_001366145.2:c.3004G>A, p.(Val1002Leu) variant. Parental analysis confirmed that the variant was absent in both parents and occurred de novo. Parental status was confirmed using 16 polymorphic markers.

Clinical genetic services and a genetic testing were done as part of routine clinical care. Written informed consent was obtained from the parents of the probands for molecular genetic analysis and possible publication of the anonymized clinical data. The study was done in accordance with local research and ethics requirements.

## Patient 4

Patient 4 is the second child of healthy parents and has a healthy brother. Pregnancy was notable by the finding of clubfeet on fetal ultrasound. Amniocentesis was performed, and karyotype was normal. Delivery was normal at 40 weeks of gestation, and birth parameters were normal. At 3 months, parents worried about a lack of visual pursuit. Complete ophthalmological examination (including fundus, ERG) was normal. The child was hypotonic and was not able to hold her head at 18 months. At 2 ½ years, she had growth restriction (weight and height at −2.5 SD) and secondary microcephaly (HC −2.5 SD). She was still unable to sit, but her tone improved slightly. She could not grab but was able to hold the toy placed in contact with her hand. She had poor visual contact. She babbled but was unable to pronounce words. She had no spasticity. She had no clinical seizures and normal EEG at this time. Control with a 24 hr EEG recorded at 4 years old showed discharges of bi-centro-parietal spikes during wake and sleep, without electrical status epilepticus during slow-wave sleep. There was no obvious motor or behavioral modification, but sometimes apneas occurred at beginning of the discharges, making clinical seizures possible. Primidone was recently started; however, at the moment, we do not have enough hindsight to judge the effectiveness of primidone. MRI performed at 1 year and 4 months showed a small vermis with slight atrophy, atrophy of cerebellar hemispheres, and thin brainstem with small protuberance (*Figure 2*). Array CGH was normal. Analysis of our NGS targeted congenital ataxia panel including *TRPM3* (panel designed after the diagnosis made in patient 1 using WES), identified the NM_001366145.2:c.3005T>G; p.(Val1002Gly) variant. Parental analysis confirmed that the variant was absent in both parents and occurred de novo. Parental status was confirmed using 16 polymorphic markers.

Clinical genetic services and a genetic testing were done as part of routine clinical care. Written informed consent was obtained from the parents of the probands for molecular genetic analysis and possible publication of the anonymized clinical data. The study was done in accordance with local research and ethics requirements.

## Patient 5

This boy is the son of patient 6. He was referred to Clinical Genetics at the age of 8 years with a history of GDD, nocturnal epilepsy, and voracious appetite. He was born at term weighing 3.09 kg following a normal pregnancy complicated only by mild pre-eclampsia from 32 weeks gestation and a maternal history of factor V Leiden requiring enoxaparin injections. His motor milestones were delayed, sitting unaided at 12 months and walking at 20 months. He required speech and language therapy from three years of age. His behavior was noted to change at around 13 months, becoming easily unsettled when he had previously been a placid baby. He developed food-seeking behavior with lack of satiety, leading to obesity. He was noted to have small genitalia and subsequently had delayed puberty with reduced testosterone levels. He has never shown any aggressive behavior and has never had regression of skills. He initially attended a mainstream school with support but subsequently transferred to special needs education. Seizures were first noted at the age of 7 years and were only present during sleep, occurring two to three times a month and gradually reducing with sodium valproate treatment, which was discontinued at the age of 14 years. Wake EEG at 7 years showed bifrontal synchronous spike and waves discharges suggestive of epileptic activity. His last seizure was at the age of 18 years. Aged 21 years, he has moderate learning difficulties and attends a college for learning life skills. He has some pain and heat insensitivity. He has had unilateral Perthes' disease, for which he is awaiting

hip replacement surgery. MRI brain did not identify any gross structural abnormalities. He had normal Fragile-X syndrome testing, normal Prader-Willi syndrome methylation analysis, and a normal karyotype. He additionally had normal methylation testing for chromosome 14 uniparental disomy. Array CGH found a maternally inherited (NCBI Build 36) 6p22.3 (18155949_18237422)×3 duplication of between 81 and 252 kb, which was not thought to account for his phenotype. Whole-exome analysis via the DDD project identified the *TRPM3* NM_001366145.2:c.3019G>A, p.(Gly1007Ser), heterozygous variant inherited from his father.

The patient was identified via the DDD study, which was granted by the UK ethical approval by the Cambridge South Research Ethics Committee (10/H0305/83).

## Patient 6

This man was the father of patient 5 and was referred to Clinical Genetics for investigation of his mild learning difficulties. He was born at 42 weeks gestation weighing 3.36 kg following a normal pregnancy. His mother had required codeine analgesia for a tooth abscess early in pregnancy. There were no specific concerns regarding his motor development, although he may have sat unaided later than average and never crawled. He took his first steps at the age of1 year and developed language at a normal time. He did not display any food-seeking behavior, aggression, or regression of skills. He attended mainstream junior school but transferred to a special educational needs senior school. He has no history of seizures. He is in paid employment, undertaking manual work in a warehouse. Heat and pain insensitivity has not been formally assessed, but the patient is reported not to feel hot when wearing warm clothing in summer time and also has a history of picking at his toenails causing traumatic dystrophy without reporting this to be painful.

He had normal testing for Fragile X syndrome and a normal array CGH. Trio whole-genome sequencing identified the de novo TRPM3 variant NM_001366145.2:c.3019G>A, p.(Gly1007Ser).

This patient was identified through diagnostic testing as part of their routine clinical care within the UK National Health Service and so no specific institutional ethical approval was required.

## Patient 7

She is the only child of unrelated parents. During the pregnancy, her mother was treated by antiretrovirals because of an HIV infection. Delivery and birth parameters were normal (3080 g, 47.5 cm, and 34 cm). The first concern was poor visual contact noted in the first weeks and later, a nystagmus and a delayed motor development. She was able to walk unaided at 20 months. Language development was only slightly delayed, and her cognitive level was in normal-low range. Audiometry was normal, and EEG was not performed in this context. At 6 years old, she was able to read but had difficulties writing and with fine motor skills in general. She was unable to climb or jump like other children. At examination, she had a mild ataxia, only detectable when she walked following a line on the floor, mild dysmetria, and nystagmus. She also had synkinesis and a prognathism. She attended mainstream school with support measures (personal school assistant, logico-mathematical reeducation, and psychomotricity). MRI performed at 2 years and 12 years showed a mild and localized atrophy of the cerebellar hemispheres (*Figure 2—figure supplement 1*).

At 16 years, weight is 48.6 kg, height 157 cm, and OFC 54 cm. Walk has drastically improved, and she has a slight nystagmus. She is in good health but had a recurrent patellar dislocation leading to the consideration of surgery. She is a special unit for inclusive education in a normal high school and attends a professional training in commerce.

NGS targeted 'cerebellar anomalies' panel identified the NM_001366145.2:c.3019G>A, p.(Gly1007Ser) missense variant in *TRPM3*. Parental analysis showed that the variant was de novo. Parental status was confirmed using 16 polymorphic markers.

Clinical genetic services and a genetic testing were done as part of routine clinical care. Written informed consent was obtained from the parents of the probands for molecular genetic analysis and possible publication of the anonymized clinical data. The study was done in accordance with local research and ethics requirements.

## Patient 8

The patient is a 4-year-old female with GDD who was born to a 32-year-old G1 P0-1 Ab0 woman at 39 weeks by cesarean section due to breech position. Her birth weight was 3.57 kg. Pre-natal period

was normal. Peri-natal period was complicated with feeding difficulties, and she remained in the hospital for 3 days. She continued with feeding difficulties and gaining weight until 8 months of age. Her formula was changed to an amino acid base, and she was diagnosed with GERD and successfully treated with lansoprazole and nizatidine.

Her development was noted to be delayed at 3 months. She was hypotonic and had no visual tracking. Her ERG was abnormal, and she was diagnosed with cortical visual impairment, strabismus, and nystagmus. She wore corrective lenses. Her brain MRI and chromosome microarray were normal. Her development has been slow, but she made progress. She can grab things with her hands, but her fine motor skills are poor, and she is unable to feed herself. She is unable to walk, but she can bear weight on her legs. She is non-verbal but uses picture cards to indicate choices and can recognize more than 20 images. She has not any seizures. Her EEGs have showed generalized background slowing, without epileptiform discharges (routine EEG and 26 hr video EEG). For the past year or two, parents have noted episodic fluctuation in her behavior. She might be very happy and playful for a week or two and then go into a period where she is irritable, crying, and is inconsolable as if in pain. With these periods of extreme irritability, there is often transient regression in her development. She might achieve things like standing, stepping with support, chewing her food, or learning to use a spoon and then she will stop doing these things for a while. Some of the episodes have been shown to be concomitant with an infection. She has no dysmorphic features, cardiac, or pulmonary problems. MRI showed only bilateral, symmetric, posterior periventricular non-specific white matter FLAIR hyperintensities.

WES identified a de novo, heterozygous, missense variant in *TRPM3* gene (NM_001366145.2:c.3376A>G; p.[Asn1126Asp]) and a de novo, heterozygous, missense variant in *PRPH2* c. 659 G>A, p.Arg220Gln pathogenic for a recessive retinopathy and possibly responsible for a macular phenotype when heterozygous.

Informed consent for participation was obtained from subjects themselves or, where necessary, their parents. The study was completed per protocol in accordance with the Declaration of Helsinki with local approval by the CHOP IRB (15–12226).

## Patient 9

Patient 9 is the oldest child of healthy, nonconsanguineous parents of northern European descent and was delivered by an uncomplicated cesarean section for breech presentation at 37 weeks' gestation. Pregnancy was complicated by pre-eclampsia. She was diagnosed with bilateral hip dysplasia and was in a Pavlik harness for 3 months. Shortly after birth, she was noted to have diffuse hypotonia, hypoactive reflexes, and roving eye movements. Difficulty tracking visual stimuli was noted at about 4 weeks, and ophthalmological evaluation revealed decreased visual acuity, mild bilateral macular pigmentary changes, normal refractive indices, bilateral ptosis, and disconjugate nystagmus. By the age of 25 months, she was demonstrating few motor movements, impaired upgaze, and increased lower extremity tone. She began receiving developmental services at the age of 3 months. At the age of 10 years, she can sit unsupported, but she cannot sit independently. She can roll over. She cannot walk and does not regularly stand with support. She can reach and grab for items of interest. She can fixate on objects visually and can track them but has been diagnosed with cortical visual impairment. She can communicate with picture cards. She has a Toby device for communication but is not proficient with this. There are some behavioral outbursts. She was diagnosed with autism spectrum disorder at the age of 8 years.

She initially had interruptible staring spells and episodes of lip smacking and hand wringing shortly after birth. She was evaluated with EEG, including a 24 hr EEG at that time; the results of which were reportedly normal. She has not had any definite clinical seizures. Her parents deny any movements suspicious for seizure, including stiffening, shaking, or staring spells. Parents also deny any repetitive, purposeless movements such as hand wringing.

On examination, she has facial dysmorphism including a short philtrum, wide nasal bridge, bulbous nose tip, and epicanthal folds. She is fed via g-tube. She is capable of visual fixation and tracking briefly and is averse to a bright light shone in her eyes. She can also blink to threat bilaterally. She has intermittent, subtle, high-frequency lateral nystagmus in primary gaze and in all directions of gaze. Her muscle bulk is diffusely decreased, and she is globally hypotonic. She is unable to stand or walk. She has normal deep tendon reflexes.

MRI scans over time have demonstrated stable periventricular leukomalacia. WES demonstrated a de novo pathogenic variant in *TRPM3* (NM_001366145.2:c.3376A>G; p.Asn1126Asp). Exome sequencing also revealed a single de novo loss of function (LOF) variant (c.2659dupA; p.R887Kf*42) in *TUBGCP5,* a gene tolerant to LOF (pLi = 0).

The participating family signed the IRB research protocol of the University of Pennsylvania division of Neurology.

## Patient 10

The patient is a 4-year-old girl, the third child of non-consanguineous German parents. The family history was unremarkable. She was born at 40th week of gestation with a birth length of 50 cm and a birth weight of 3430 g. Her psychomotor development was delayed. At age of 12 months, she was able to sit, and at age of 24 months, she started to walk. At the age of 24 months, she began to speak. At the age of 2.5 years, EEG investigations indicated electrical status ESES. Because of the diagnosis (TRPM3 gain-of-function mutation) and the literature showing primidone as an antagonist of TRPM3, a treatment by primidone was initiated. At this time, she had ataxia that improved with primidone treatment as well as EEG.

Physical examination at age of 3 years showed a height of 94 cm (+0.05 SD), a weight of 19 kg (+1.81 SD), and an HC of 51.5 cm (+1.53 SD).

Her facial signs included a flat midface, a flat and broad nasal bridge with a broad nasal tip, upward slanting palpebral fissures, strabismus, and full lips. Her fingers were short and the metacarpal bones IV and V also appeared to be short. Her toes II to V also showed brachydactyly. Cranial MRI at the age of 3 years was normal. In addition, she had immune thrombocytopenia and hypochromic microcytic anemia.

Cytogenetic analysis and molecular karyotyping gave normal results. Trio exome sequencing identified a de novo variant in *TRPM3:* NM_001366145.2:c.3397T>C, p.(Ser1133Pro).

The participating family, consisting of the mother, father, and female proband, provided written consent and was enrolled into the C4RCD research protocol at the TGen. Written consent for the proband under the age of 18 years was obtained from the parents. The study protocol and consent documents were approved by the WIRB (# 20120789). The retrospective analysis of epilepsy patient data was approved by the local ethics committees of the Charité (approval no. EA2/084/18).

## Comment on the numbering of *TRPM3* variants and DNA constructs

The human *TRPM3* gene contains 28 exons, and alternative splicing of the primary transcripts gives rise to a large number of splice isoforms, leading to ambiguity in the numbering of gene variants. Most of the previous reports based the numbering of disease-associated variants on the NM_020952.4 reference sequence (*de Sainte Agathe et al., 2020*; *Dyment et al., 2019*; *Gauthier et al., 2021*; *Lines et al., 2022*), including the Human Gene Mutation Database (*Stenson et al., 2017*) and OMIM (https://www.omim.org/), whereas NM_001007471.2 was used as reference by others (*Van Hoeymissen et al., 2020*; *Zhao et al., 2020*). In this report, we based the numbering of the TRPM3 variants on the Mane transcript NM_001366145.2 sequence (see explanation in the section 'Organization of the human *TRPM3* gene and alternative splicing in the cerebellum' in Results).

For functional expression, we used four different WT TRPM3 constructs, representing different splice isoforms (*Figure 1—figure supplement 1*; *Table 2*; *Vriens et al., 2011*; *Zhao et al., 2020*). For single-cell calcium imaging, we used the isoform corresponding to GenBank AJ505026.1, with

**Table 2.** Overview used splice isoforms.

| Splice isoform | Start | Exon 8 | Exon 15 | Exon 17 | Exon 24 | Exon 28 | Functionality* |
|---|---|---|---|---|---|---|---|
| AJ505026.1 | Exon 2 | − | − | + | Short | Spliced | Normal |
| NM_001366141.2 | Exon 1 | − | − | + | Short | Full | Normal |
| NM_001366145.2 | Exon 2 | − | + | + | Short | Full | Normal |
| NM_001366147.2 | Exon 2 | + | + | + | Short | Full | No activity |

*Normal functionality refers to whole-cell currents activated by ligands including pregnenolone sulfate (PS), CIM0216, and clotrimazole, with biophysical properties as described in *Held et al., 2015* and *Vriens et al., 2014*.

**Table 3.** Characteristics of the variants.

| Variant | gnomAD | SIFT | CADD | PROVEAN | DANN |
|---|---|---|---|---|---|
| c.1841A>T p.(Asp614Val) | Absent | Deleterious | 25.3 | Damaging | 0.9865 |
| c.2305C>G p.(Leu769Val) | Absent | Tolerated | 25.9 | Damaging | 0.9986 |
| c.3004G>T p.(Val1002Leu) | Absent | Tolerated | 24.8 | Damaging | 0.9969 |
| c.3005T>G p.(Val1002Gly) | Absent | Deleterious | 26.3 | Damaging | 0.9973 |
| c.3019G>A p.(Gly1007Ser) | Absent | Deleterious | 27.8 | Damaging | 0.9986 |
| c.3376A>G p.(Asn1126Asp) | Absent | Tolerated | 28.4 | Damaging | 0.9977 |
| c.3397T>C p.(Ser1133Pro) | Absent | Deleterious | 27.2 | Damaging | 0.9987 |

Characteristics of the variants (database, predicted pathogenicity). Overview of the different variants and the identification protocol.

YFP directly linked to the channel's C-terminus, cloned in the pCDNA3 vector. For plate reader-based experiments, we used the isoform corresponding to NCBI reference sequence NM_001366141.2 cloned in the pcDNA3.1(+)-N-eGFP vector. For whole-cell patch-clamp experiments, we used (a) the isoform corresponding to GenBank AJ505026.1, (b) the isoform corresponding to NCBI reference sequence NM_001366145.2, and (c) the isoform corresponding to NCBI reference sequence NM_001366147.2, both (b) and (c) were cloned in the pCAGGSM2_Ires_GFP vector. Human disease-associated variants were introduced using the standard PCR overlap extension method, and variant sequences were verified by sequencing of the entire DNA constructs (*Vriens et al., 2007*). As indicated in the results section, variant numbering was based on the amino acid position of the mutated residue in the NM_001366145.2 isoform. According to this numbering, the recurrent variant p.Val1027Met will be referred to as p.Val1002Met, p.Pro1127Gln as p.Pro1102Gln, and p.Ser1392Thr as p.Ser1367Thr (*Table 3*).

## Sequence alignment

CLUSTAL Omega (1.2.4) multiple sequence alignment was used to perform the alignment and to obtain the Phylogenetic Tree. Jalview (2.11.2.4) was used for visualization.

## Cell culture and transfection

HEK293T cells (identifier ATCCCRL-3216) were kindly provided by Dr. S Roper (University of Miami school of medicine Department of physiology and biophysics, 4044 Miami FL 33136). The cells were cultured as described previously (*Vriens et al., 2007*) and used up to passage number 25. The cells were tested for the lack of mycoplasma. For patch-clamp and single-cell calcium imaging, cells were transfected with 2 µg of channel cDNA using TransIT-293 transfection reagent (MirusBio) (*Vriens et al., 2007*) and analyzed 36–48 hr after transfection. For the intracellular $Ca^{2+}$ measurements using a fluorescent microplate reader, cells were transfected with 400 ng of channel cDNA plus 1 µg of GCaMP6 using Effectene transfection reagent (Qiagen). When indicated, to mimic heterozygous conditions, a mixture of WT and *TRPM3* variant cDNA was used (ratio 1:1). NT HEK293T cells were used as negative controls in all experiments.

## Calcium microfluorimetry

The imaging system for standard single cell calcium measurements has been described before (*Vriens et al., 2011*). Briefly, cells were incubated with 2 µM Fura-2-acetoxymethyl ester (Thermo Fisher Scientific) in 1 ml culture medium for 20–60 min at 37°C. Fluorescent signals evoked during alternating illumination at 340 and 380 nm using a Lambda XL illuminator (Sutter Instrument, Novato, CA, USA) and recorded by an Orca Flash 4.0 camara (Hamamatsu Photonics, Belgium) on a Nikon Eclipse Ti fluorescence microscope (Nikon Benelux, Brussels, Belgium). The imaging data were recorded using NIS-element software (NIS-Elements). Absolute calcium concentrations were calculated from the ratio of the fluorescence signals at both wavelengths ($F_{340}/F_{380}$) after correction for the individual background fluorescence signals, using the Grynkiewicz equation (*Grynkiewicz et al., 1985*): $[Ca^{2+}]=K_m \times$

$(R-R_{min})/(R_{max}-R)$, where $K_m$, $R_{min}$, and $R_{max}$ were estimated from in vitro calibration experiments with known calcium concentrations. The standard imaging solution contained (in mM): 150 NaCl, 2 CaCl$_2$, 1 MgCl$_2$, and 10 HEPES, pH 7.4 with NaOH (~320 mOsm). Calcium amplitudes were calculated as the difference between the maximum calcium concentration during the period of stimulus application and the basal value before stimulation of responding cells. At the start of the recording, cellular YFP fluorescence was determined as a measure of TRPM3 protein expression levels. All data represent the results from at least three independent coverslips.

For the intracellular Ca$^{2+}$ measurements using a fluorescent microplate reader, HEK293T cells were plated on poly-D-lysine coated black-wall clear-bottom 96-well plates after 24 hr of transfection, and measurements were performed 24–48 hr after plating. Experiments were performed at room temperature in a buffer containing (in mM): 137 NaCl, 5 KCl, 1 MgCl$_2$, 2 CaCl$_2$, 10 HEPES, and 10 glucose (pH 7.4) with NaOH. Intracellular Ca$^{2+}$ levels were measured by a Flexstation-3 96-well plate reader (Molecular Devices). GCaMP6 signal was detected at excitation wavelengths 485 nm, and fluorescence emission was detected at 525 nm. Various concentrations of PS were applied to activate TRPM3 channels, and 2 µM ionomycin was applied at the end of the experiment to induce the maximum calcium influx. For every experimental group, three transfections were performed, and signals from four replicates were collected for different PS concentrations within the same transfection.

## Electrophysiology

Whole-cell patch-clamp recordings were performed using an EPC-10 amplifier and the PatchMasterPro software (HEKA Elektronik). Current measurements were done at a sampling rate of 20 kHz, and currents were digitally filtered at 2.9 kHz. In all measurements, 70% of the series resistance was compensated. The standard internal solution contained (in mM): 100 CsAsp, 45 CsCl, 10 EGTA, 10 Hepes, and 1 MgCl$_2$ (pH 7.2 with CsOH); the standard extracellular solution contained (in mM): 150 NaCl, 1 MgCl$_2$, and 10 Hepes (pH 7.4 with NaOH). The standard patch pipette resistance was between 2 and 4 MΩ when filled with pipette solution. All experiments were performed at room temperature (23 ± 1°C). To evaluate the channel activity at basal level and in the presence of PS (40 µM), a voltage step protocol was applied in which voltage steps of +40 mV were applied starting from –160 mV toward +160 mV with a holding potential at 0 mV.

## Chemicals

All chemicals were obtained from Sigma-Aldrich, ionomycin was purchased from Cayman Chemical. PS and primidone were dissolved in the bath solutions from a 100 mM stock diluted in Dimethyl sulfoxide (DMSO).

## Statistics

Calcium microfluorimetry data were analyzed with NIS-Elements software (Nikon, Japan), Excel (Microsoft, WA, USA), IgorPro 6.2 (WaveMetrics, OR, USA), and OriginPro 9.5 (OriginLab, MA, USA). RStudio Team (2020; Integrated Development for R. RStudio, PBC, Boston, MA, USA; URL http://www.rstudio.com/), GraphPad Prism (9.2.0) and OriginPro 9.5 were further used for statistical analysis and data display. All data sets were tested for normality using the Shapiro-Wilk test and depending on the outcome, a one-way ANOVA or a Kruskal-Wallis test with subsequent Tukey's, Dunn's, or Dunnett's posthoc tests were performed, or a two-way ANOVA with Sidak's multiple comparison was used. When having paired non-normally distributed data sets, a Wilcoxon signed rank test was performed. p-Values below 0.05 were considered as significant. Data points represent means ± SEM of the given number (n) of identical experiments. No exclusion of statistical outliers was performed in this study.

## Acknowledgements

We thank all parents and children for their willingness to participate to this study. We thank all members of the Laboratory of Endometrium, Endometriosis and Reproductive Medicine (LEERM) and the members of the Laboratory of Ion Channel Research (LICR) Leuven for their help during experiments and useful discussions. We thank Dr. L Gaspers for the kind gift on GCaMP6, and Bahar Bazeli for help with visualizing the TRPM3 variants on the cryo-EM structure. The study was supported by the Einstein Stiftung Fellowship through the Günter Endres Fond.

We thank the Research Foundation-Flanders FWO, (G.0D1417N, G.084515N, G.0A6719N), the Research Council of the KU Leuven (C14/18/106, C3/21/049) for funding the project of JV, Einstein Stiftung for AMK, and the association "Connaître les syndromes cérébelleux" for funding the project of LB. EVH is a fellow of the Research Foundation-Flanders FWO (11E782N).

## Additional information

### Funding

| Funder | Grant reference number | Author |
| --- | --- | --- |
| FWO | G.0D1417N | Joris Vriens |
| FWO | G.084515N | Joris Vriens |
| FWO | G.0A6719N | Joris Vriens |
| FWO | 11E782 | Evelien Van Hoeymissen |
| Queen Elisabeth Medical Foundation for Neurosciences | | Thomas Voets |
| Vlaams Instituut voor Biotechnologie | | Joris Vriens |
| FWO | G079623N | Joris Vriens |

The funders had no role in study design, data collection and interpretation, or the decision to submit the work for publication.

### Author contributions

Lydie Burglen, Conceptualization, Resources, Writing – original draft; Evelien Van Hoeymissen, Data curation, Formal analysis, Methodology, Visualization, Writing – review and editing; Leila Qebibo, Magalie Barth, Newell Belnap, Felix Boschann, Christel Depienne, Andrew GL Douglas, Mark P Fitzgerald, Nicola Foulds, Catherine Garel, Ingo Helbig, Denise Horn, Vinodh Narayanan, Mailys Rupin-Mas, Alexandra Afenjar, Vincent Th Ramaekers, Sarah M Ruggiero, Simon Thomas, Stéphanie Valence, Lionel Van Maldergem, Diana Rodriguez, Data curation; Katrien De Clercq, Siyuan Zhao, Data curation, Visualization; Katharina Held, Conceptualization; Annelies Janssen, Methodology; Angela M Kaindl, Writing – original draft; Christina Prager, Conceptualization, Supervision; Tibor Rohacs, Supervision; David Dyment, Thomas Voets, Conceptualization, Supervision, Writing – original draft; Joris Vriens, Supervision, Funding acquisition, Investigation, Visualization, Methodology, Writing – original draft

### Author ORCIDs

Evelien Van Hoeymissen (ID) http://orcid.org/0000-0003-3897-8998
Denise Horn (ID) http://orcid.org/0000-0003-0870-8911
Annelies Janssen (ID) http://orcid.org/0000-0002-6735-8248
Angela M Kaindl (ID) http://orcid.org/0000-0001-9454-206X
Vinodh Narayanan (ID) http://orcid.org/0000-0002-0658-3847
Siyuan Zhao (ID) http://orcid.org/0000-0003-2005-9440
Tibor Rohacs (ID) http://orcid.org/0000-0003-3580-2575
Joris Vriens (ID) http://orcid.org/0000-0002-2502-0409

### Ethics

The study was performed in accordance with the guidelines specified by the institutional review boards and ethics committees at each institution. Information of institutional protocols are provided in the section of Material & Methods. All parents agreed on sharing and publicing the patients' information. Patients information: Patient 1, 3, 4 and 7: Written informed consent was obtained from the parents of the probands for molecular genetic analysis and possible publication of the anonymized clinical data. The study was done in accordance with local research and ethics requirements. Patient 2: Parents signed an informed consent, received a genetic counselling before and after the analysis,

and the genetic study was performed in accordance with German and French ethical requirements and laws. Patient 5: UK ethical approval by the Cambridge South Research Ethics Committee (10/H0305/83). Patient 6: outine clinical care within the UK National Health Service, and so no specific institutional ethical approval was required. Patient 8: Declaration of Helsinki with local approval by the Children's Hospital of Philadelphia (CHOP) Institutional Review Board (IRB 15-12226). Patient 9: The participating family signed the IRB research protocol of the University of Pennsylvania division of Neurology. Patient 10: The study protocol and consent documents were approved by the Western Institutional Review Board (WIRB # 20120789). The retrospective analysis of epilepsy patient data was approved by the local ethics committees of the Charité (approval no. EA2/084/18).

### Decision letter and Author response
Decision letter https://doi.org/10.7554/eLife.81032.sa1
Author response https://doi.org/10.7554/eLife.81032.sa2

---

## Additional files

### Supplementary files
• MDAR checklist

### Data availability
Raw data for the following figures are made available via figshare (https://doi.org/10.6084/m9.figshare.21799604): Figure 1 - Figure Supplement 1; Figure 3; Figure 3 - Figure Supplement 1, 2, 3 and 4; Figure 4; Figure 4 - Figure Supplement 1, 2 and 3.

The following dataset was generated:

| Author(s) | Year | Dataset title | Dataset URL | Database and Identifier |
|---|---|---|---|---|
| Van Hoeymissen E, Rohacs T | 2022 | Data of Evelien Van Hoeymissen for manuscript "Gain-of-function variants in the ion channel gene TRPM3 underlie a spectrum of neurodevelopmental disorders" | https://doi.org/10.6084/m9.figshare.21799604 | figshare, 10.6084/m9.figshare.21799604 |

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
