## [Editor Report]

This important study is of interest to scientists and clinicians working to understand genetic causes of neurodevelopmental disability, cerebellar ataxia, and epilepsy. It represents the most comprehensive functional characterization of disease-causing mutations in the TRPM3 gene and includes a discussion of novel cerebellar signs and symptoms affecting a subset of affected individuals. Using calcium imaging and electrophysiology, solid evidence is provided that disease-causing mutations have a gain-of-function phenotype when expressed together with WT subunits, as in the patients in the study who are heterozygous for the mutations.

---

## [Decision Letter]

**Decision letter after peer review:**

Thank you for submitting your article "Gain-of-function variants in the ion channel gene TRPM3 underlie a spectrum of neurodevelopmental disorders" for consideration by *eLife*. Your article has been reviewed by 3 peer reviewers, including Andrés Jara-Oseguera as the Reviewing Editor and Reviewer #1, and the evaluation has been overseen by Richard Aldrich as the Senior Editor.

The reviewers have discussed their reviews with one another, and the Reviewing Editor has drafted this to help you prepare a revised submission. All three reviewers were in agreement that this is a carefully done manuscript that makes important contributions that will be of interest to both clinical scientists and ion channel biophysicists. However, the reviewers raised some concerns that should be addressed before publication, which are summarized below.

Essential revisions:

1) Differences in exon composition and amino acid sequence should be clearly described for each of the five constructs that are examined in the study, both in Figure 1 and in each of the figure legends. A justification should be provided for using different background constructs to study the mutations.

2) A more detailed characterization of the mutants is required, at least in regards to their surface expression in the plasma membrane. Surface biotinylation followed by pull-down and western blotting, or using a membrane-impermeable fluorescent dye directed at the channels would be appropriate. Alternatively, the current densities of the different constructs in a sufficiently large number of cells should be evaluated using patch-clamp electrophysiology – patch-clamp experiments should be performed for at least L769V, G1007S channels expressed alone and together with WT, showing the extent of baseline channel activity, and the magnitude in the response to PS.

3) Individual data points should be included in Figure 3B, D and Figure 4B, D. For the (B) panels, data from individual cells at each of the two conditions could be linked by a line to allow readers to assess the robustness of the findings and the cell-to-cell variability. I suggest displaying the data in the (D) panels in a similar fashion, which would be more informative than how data is currently displayed.

4) A sequence alignment should be included that allows more precise and mechanistically informative mapping of the mutants onto the channel sequence. The alignment could include TRPM3 orthologues from other species, as well as other TRPM channels that are closely related and whose structures have been determined. A discussion on the relative conservation at these positions would be informative.

5) Please comment or discuss the use of such high concentrations of primidone (100 µM) when the reported KD for its block of TRPM3 is ~1µM (Krugel et al., Pain, 2017).

6) Please comment or discuss whether mutations in the upstream genetic components (receptor, Gbeta, Ggamma, or prenylation enzymes) are coincident with any of the phenotypes.

7) Patient 1: authors refer to "kinetic and static cerebellar symptoms" at 365, please specify. Kinetic and static are often dichotomous pairs used to describe tremor (is it a kinetic and static tremor that the authors are reporting here).

– Patient 1: also at 365, "ocular saccadic pursuit" is written. Is the meaning that the patient had saccadic intrusions? Dysmetric saccades? Or saccadic breakdown of smooth pursuit? The table refers to nystagmus…

8) Patient 2: line 375 says "diffuse retinal dysfunction." How was the retinal dysfunction diagnosed? Did the patient have ERG? Was it a structural finding?

9) Patient 4: line 424 "Seizures were doubtful…apneas at the beginning of discharges" is unclear. Apnea can be a clinical manifestation of seizures. A more detailed explanation is required.

10) Patient 5: Line 451 and table "spike and waves." Were these epileptiform findings generalized or focal and, if the latter, where were they located?

11) Patient 9: "nystagmoid" is used at 541 and 557. Please clarify.

12) Patient 10: CSWS is referred to in the text and ESES in the table. Use one or the other and indicate in the description of the patient whether the finding was felt to be strictly an EEG pattern or whether it was felt to be part of epileptic encephalopathy. Primidone was given, presumably directed by the genetic results: was it effective clinically and/or electrographically?

13) Please give ages that EEGs were performed for all patients who had EEGs and whether wake and sleep were recorded for each

14) Additional information or discussion should be provided about the behavior of TRPM3 channels in neurons, and ideally in neurons of the cerebellum. Other information would also be useful, is known, such as the types of neurons in the cerebellum of the brain in general that express TRPM3 channels.

---

## [Author Response]

Essential revisions:1) Differences in exon composition and amino acid sequence should be clearly described for each of the five constructs that are examined in the study, both in Figure 1 and in each of the figure legends. A justification should be provided for using different background constructs to study the mutations.

We thank you for this useful suggestion and we agree that additional information is required to clarify the figure legends. Information on the used isoforms are now provided in each of the figure legends. In addition, a short motivation of the different isoforms is now included in the section of Material & Methods. In addition, a additional table (Table 2) in included in the revised version of the manuscript to further clarify the different constructs.

2) A more detailed characterization of the mutants is required, at least in regards to their surface expression in the plasma membrane. Surface biotinylation followed by pull-down and western blotting, or using a membrane-impermeable fluorescent dye directed at the channels would be appropriate. Alternatively, the current densities of the different constructs in a sufficiently large number of cells should be evaluated using patch-clamp electrophysiology – patch-clamp experiments should be performed for at least L769V, G1007S channels expressed alone and together with WT, showing the extent of baseline channel activity, and the magnitude in the response to PS.

To further investigate the expression of the TRPM3 variants at the plasma membrane, we evaluated L769V and G1007S mutants in whole-cell patch-clamp experiments in both homozygous and heterozygous (WT + mutant) conditions. These additional results are now included in figure 4Supplementary figure 3 of the revised manuscript. In combination with the YFP fluorescent measurements, they provide further support for our conclusions that the mutations lead to increased channel activity. A detailed study of potential effects on channel trafficking and stability at the plasma membrane of all the different disease-associated variants, with and without co-expression of WT TRPM3, is ongoing, but falls beyond the scope of the present study.

3) Individual data points should be included in Figure 3B, D and Figure 4B, D. For the (B) panels, data from individual cells at each of the two conditions could be linked by a line to allow readers to assess the robustness of the findings and the cell-to-cell variability. I suggest displaying the data in the (D) panels in a similar fashion, which would be more informative than how data is currently displayed.

We thank the reviewers for this suggestion. We agree that adding individual datapoints is more informative as it allows the reader to assess the robustness of our data. However, since there are many data points per condition, plotting the individual datapoints over the bar plots will make the figure rather complicated and unclear. Therefore, we added supplementary figures (Figure 4 —figure supplement 1, 2), displaying the individual data points.

4) A sequence alignment should be included that allows more precise and mechanistically informative mapping of the mutants onto the channel sequence. The alignment could include TRPM3 orthologues from other species, as well as other TRPM channels that are closely related and whose structures have been determined. A discussion on the relative conservation at these positions would be informative.

We have followed the suggestion of the reviewers and included a sequence alignment, including TRPM3 orthologues from other species as well as other members of the TRPM family. This additional information convincingly showed the conservation of the amino acid residues in TRPM3 orthologues from other species (*Drosophila*, Zebrafish, mouse, rat, macaca). The alignment with other members of the TRPM subfamily illustrates that all of the residues are conserved in the close homologues of TRPM3 namely TRPM1, TRPM6 and TRPM7. In other non-closely related TRPM members (TRPM4, TRPM5, TRPM2 and TRPM8) some residues are no longer present and substituted by other residues. This additional information is included in the additional Figure 1 —figure supplement 3. Moreover, we now included a figure mapping the residues on the very recent TRPM3 cryo-EM structure and additional discussion, which provides further mechanistically relevant information.

5) Please comment or discuss the use of such high concentrations of primidone (100 µM) when the reported KD for its block of TRPM3 is ~1µM (Krugel et al., Pain, 2017).

We acknowledge that we used a very high concentration of primidone compared to the described IC50 value. The main motivation for this is that some of the mutants were less sensitive for primidone compared to wild-type TRPM3, as published in our earlier work (Van Hoeymissen et al., *eLife*, 2020) and work of others (Zhao S et al., 2020, *eLife*). Primidone was used to illustrate that the elevated basal cytosolic calcium concentrations were caused by the basal overactivity of the variants. References to this work is now included in the revised version of the manuscript.

6) Please comment or discuss whether mutations in the upstream genetic components (receptor, Gbeta, Ggamma, or prenylation enzymes) are coincident with any of the phenotypes.

We included a short paragraph in the discussion where we comment on the possible link between genetic variant in TRPM3 and previously described upstream genetic components. Indeed, there are some known mutations in the GNB1^1^, GNB2^2^ and GNB5^3^ gene causing similar phenotypes as the patients described in this work.

Petrovski S, Kury S, Myers CT, Anyane-Yeboa K, Cogne B, Bialer M, et al. Germline de novo Mutations in GNB1 Cause Severe Neurodevelopmental Disability, Hypotonia, and Seizures. Am J Hum Genet. 2016;98(5):1001-10.

Fukuda T, Hiraide T, Yamoto K, Nakashima M, Kawai T, Yanagi K, et al. Exome reports A de novo GNB2 variant associated with global developmental delay, intellectual disability, and dysmorphic features. Eur J Med Genet. 2020;63(4):103804.

Lodder EM, De Nittis P, Koopman CD, Wiszniewski W, Moura de Souza CF, Lahrouchi N, et al. GNB5 Mutations Cause an Autosomal-Recessive Multisystem Syndrome with Sinus Bradycardia and Cognitive Disability. Am J Hum Genet. 2016;99(3):704-10

7) Patient 1: authors refer to "kinetic and static cerebellar symptoms" at 365, please specify. Kinetic and static are often dichotomous pairs used to describe tremor (is it a kinetic and static tremor that the authors are reporting here).

We appreciate the comments of the referees concerning the lack of patient information. The requested information is included in the manuscript. The patient had a complete cerebellar motor syndrome: ataxia of stance, ataxic gait, dysmetria, adiadochokinesia, intention tremor, dysarthria and saccadic pursuit.

– Patient 1: also at 365, "ocular saccadic pursuit" is written. Is the meaning that the patient had saccadic intrusions? Dysmetric saccades? Or saccadic breakdown of smooth pursuit? The table refers to nystagmus…

The patient did not show eye movements during clinical observation. In young patients, only a clinical examination for abnormal eye movements is performed, and therefore it is difficult to be highly specific in the description of the abnormal ocular movements. Unlike adults, eye movements recording in young children is difficult and not routinely performed in our center, particularly in disabled children.

We performed only a clinical examination and concluded to saccadic breakdown of smooth pursuit.

This information is included in the table and the clinical description.

8) Patient 2: line 375 says "diffuse retinal dysfunction." How was the retinal dysfunction diagnosed? Did the patient have ERG? Was it a structural finding?

The patient had a severely reduced central visual acuity, photophobia, normal funduscopy and a mitigated photopic and scotopic ERG responses with normal flash evoked visual responses. These findings were compatible with a mainly central retinal dysfunction. She also had a Mittendorf cataract at the right eye. The table and clinical summary have been corrected in the revised manuscript.

9) Patient 4: line 424 "Seizures were doubtful…apneas at the beginning of discharges" is unclear. Apnea can be a clinical manifestation of seizures. A more detailed explanation is required.

We thank you for the comment, and agree with the comment that the sentence was not clear. Indeed, apnea can be a clinical manifestation of seizures and because of this doubt, the EEG anomalies, and the nature of the gene, a primidone therapy was started as noted in the table. We missed to report the therapy in the clinical summary in the submitted manuscript (noted in table). The sentence in the clinical description was corrected and clarified: There was no obvious motor or behavioral modification, but sometimes apneas occurred at beginning of the discharges, making clinical seizures possible. Primidone was started but, because the beginning of this therapy is recent in this child, we do not have enough hindsight to judge its effectiveness. We also added a sentence in the results part (line 176).

10) Patient 5: Line 451 and table "spike and waves." Were these epileptiform findings generalized or focal and, if the latter, where were they located?

The EEG showed bifrontal synchronous spike and waves discharges. There was also frequent right sided spike and wave discharges over posterior temporal/centroparietal area. This clarification is added in the clinical summary.

11) Patient 9: "nystagmoid" is used at 541 and 557. Please clarify.

The patient 9 had disconjugate nystagmus. This clarification has been corrected in the clinical summary of the manuscript.

12) Patient 10: CSWS is referred to in the text and ESES in the table. Use one or the other and indicate in the description of the patient whether the finding was felt to be strictly an EEG pattern or whether it was felt to be part of epileptic encephalopathy. Primidone was given, presumably directed by the genetic results: was it effective clinically and/or electrographically?

This was harmonized : we describe the EEG anomalies as ESES (Electrical status epilepticus during slowwave sleep) in the manuscript and in the table. It was only an EEG pattern, the patient did not have clinical seizures. Because of the *TRPM3* diagnosis, and the literature showing that primidone is an antagonist of TRPM3, primidone was started and was effective clinically on ataxia and EEG normalized*.*

This additional information has been modified in the description of patient 10 in the manuscript.

13) Please give ages that EEGs were performed for all patients who had EEGs and whether wake and sleep were recorded for each

The age and conditions of EEG have been added in the clinical summary and table, when missing.

14) Additional information or discussion should be provided about the behavior of TRPM3 channels in neurons, and ideally in neurons of the cerebellum. Other information would also be useful, is known, such as the types of neurons in the cerebellum of the brain in general that express TRPM3 channels.

We thank the reviewers for this suggestion. We agree that adding additional information on the expression of TRPM3 in the brain would be informative for the reader and therefore reanalyzed publicly available single cell RNA sequencing datasets of the developing and adult cerebellum and adult cortex. Shortly, this expression data points towards an important role of TRPM3 in specific brain tissues both in early development and adult stage. This additional information is included in the Discussion section, in Figure 5 and in the additional Figure 5 —figure supplement 1,2.